# Divisive Feature Normalization Improves Image Recognition Performance in AlexNet

**Michelle Miller**[1]**, SueYeon Chung**[1,2,3]**, Ken D. Miller**[1,4]
[1] Center for Theoretical Neuroscience, Columbia University, [2] Center for Neural Science, New York University, [3] Flatiron Institute, Simons Foundation, [4] Swartz Program in Theoretical Neuroscience, Kavli Institute for Brain Science, Department of Neuroscience, College of Physicians and Surgeons, Zuckerman Mind Brain Behavior Institute, Columbia University

## Abstract

Local divisive normalization provides a phenomenological description of many nonlinear response properties of neurons across visual cortical areas. To gain insight into the utility of this operation, we studied the effects on AlexNet of a local divisive normalization between features, with learned parameters. Developing features were arranged in a line topology, with the influence between features determined by an exponential function of the distance between them. We compared an AlexNet model with no normalization or with canonical normalizations (Batch, Group, Layer) to the same models with divisive normalization added. Divisive normalization always improved performance for models with batch or group or no normalization, generally by 1-2 percentage points, on both the CIFAR-100 and ImageNet databases. To gain insight into mechanisms underlying the improved performance, we examined several aspects of network representations. In the early layers both canonical and divisive normalizations reduced manifold capacities and increased average dimension of the individual categorical manifolds. In later layers the capacity was higher and manifold dimension lower for models roughly in order of their performance improvement. Examining the sparsity of activations across a given layer, divisive normalization layers increased sparsity, while the canonical normalization layers decreased it. Nonetheless, in the final layer, the sparseness of activity increased in the order of no normalization, divisive, combined, and canonical. We also investigated how the receptive fields (RFs) in the first convolutional layer (where RFs are most interpretable) change with normalization. Divisive normalization enhanced RF Fourier power at low wavelengths, while divisive+canonical enhanced power at mid (batch, group) or low (layer) wavelengths, compared to canonical alone or no normalization. In conclusion, divisive normalization enhances image recognition performance, most strongly when combined with canonical normalization, and in doing so it reduces manifold capacity and sparsity in early layers while increasing them in final layers, and increases low- or mid-wavelength power in the first-layer receptive fields.

## 1 Introduction

Neural networks (NN's) in general and convolutional NN's (CNN's) in particular were originally inspired by the brain. However, only the barest sketch of brain function has been incorporated into NN's. Conversely, studies of brain-like function in NN's have only begun to impact neuroscience. Here we consider a biological form of "divisive normalization" (DN), which is postulated to be a canonical computation of at least sensory cortex (Carandini & Heeger, 2012). We show that it can enhance the image classification performance of AlexNet (Krizhevsky et al., 2012), and study how it alters representations in the context of this architecture and task.

Divisive normalization is a phenomenological description (Geisler & Albrecht, 1992; Heeger, 1992) of nonlinear neuronal response properties observed throughout sensory cortex: when multiple stimuli are simultaneously presented, either within a neuron's receptive field (RF; the region of sensory space in which appropriate stimuli drive a given neuron's response) or both inside the RF (in the "center") and outside of it ("surround"), then (1) responses tend to be less than the sum of the responses to the

stimuli shown individually, that is, summation is sublinear; but (2) when stimuli are weak, summation becomes more linear or even supralinear (more than the sum of the responses to the individual stimuli). The phenomenological description posits that a neuron's response is its unnormalized response, divided by a function of a constant plus a sum over the unnormalized responses (perhaps raised to a power) of all the other surrounding neurons in a "normalization pool". Thus, anything that adds to the collective response of the population also suppresses (effectively inhibits) each individual neuron's response. However, the divisive function reduces to the constant for weak unnormalized responses, thus removing the effects of normalization for weak stimuli. The unnormalized response is often modeled as an expansive function, e.g. a rectified quadratic; then the response to multiple weak stimuli can show supralinear summation.

There are other standard forms of normalization being used in neural networks (Ren et al., 2016), which we will call "canonical". These include (but are not limited to) batch (Ioffe & Szegedy, 2015), layer (Ba et al., 2016), instance (Ulyanov et al., 2016), and group (Wu & He, 2018) normalization. These all standardize (zero mean, unit variance) and then affinely transform sets of activations in a given layer; they differ in the sets of channels and images over which standardization is performed (for all, the sets include all of space). These normalizations prevent or reduce covariate shifts and can have other advantages. The set includes one image for all but batch normalization, which uses all images in a batch. The first three do not lead to competition between channels, as the set either includes a single channel (batch or instance) or all channels (layer), and the same operations are being applied to all channels in the set. However, in group normalization, the channels are divided into non-overlapping groups, with standardization performed separately over each group. Thus there is competition within the group – one channel's strong activity can suppress the activity of other channels in the group, relative to the activities of channels in other groups. This is closest to DN, which is competitive.

In our formulation of DN, the group with which a neuron is competing changes continuously with the neuron, constituting some local region around each neuron. We take this local region to be a single point of space and a learnable span of channels about a given channel. More precisely, the channels are topologically arranged on a line; the contribution of nearby channels is weighted according to the distance between it and the channel being normalized, by a decaying exponential with a learnable length constant. DN does not prevent covariate shifts, so we will find it useful to combine DN with one of the other normalizations.

Our contributions in this paper are, for the first time (to our knowledge), characterizing how a canonical biological operation, DN, learned along with the CNN filters, affects ImageNet and CIFAR-100 performance and learned representations in a CNN (AlexNet) with and without "canonical" normalizations. In particular, we show:

- Addition of DN improves performance for image recognition in AlexNet models with or without canonical normalizations, and the best performance is found by combining both types of normalization;

- DN increases the large or medium (depending on presence and type of canonical normalization) wavelength Fourier modes in the first layer receptive fields.

- Both canonical and divisive normalizations reduce the network's manifold capacity and correspondingly change associated geometric measures at interior layers, leading to improved manifold capacity and associated changes in geometric measures at the final level, corresponding well to improvements in performance.

- DN consistently increases the sparsity of activations (Gini index) at each normalization step and in the output layer.

We also find preliminary evidence suggesting that DN can improve out-of-distribution (OOD) performance. This work should be of interest both to the ML and neuroscience communities, and warrants further study, for example, to understand why DN produces the associated changes in representations, whether and how these changes are related to the improvements in performance, and how performance with DN can be optimized.

## 2 RELATED WORK

In recent work a neural circuit model was found that produces the neural responses that had been phenomenologically described by DN, along with a number of other biological response properties (Ahmadian et al., 2013; Rubin et al., 2015). This has raised interest in understanding the possible functions of this normalization, for which there are many hypotheses, of which we mention only a few. It has been postulated to keep activations within an appropriate dynamic range for the neurons (Carandini & Heeger, 2012). It has been shown to remove higher-order statistical dependencies in responses to auditory or visual stimuli (Schwartz & Simoncelli, 2001), and more generally to minimize redundancy, maximize information, or efficiently or optimally encode (Malo & Laparra, 2010; Gomez-Villa et al., 2020; Malo, 2020; Ballé et al., 2016). It has also been shown to arise from statistical inference of the reflectances underlying a model of the statistics of natural scenes, the Gaussian scale mixture model (GSM) (Coen-Cagli et al., 2012; 2015; Echeveste et al., 2020).

The original AlexNet (Krizhevsky et al., 2012) included local response normalization (LRN), much like ours (Eq. 1) but with a linear numerator and the sum in the denominator over $\pm 2$ neighbors without exponential weighting. Parameter values were hyperparameters set using a validation set; the equivalent of our parameter $k\alpha/\lambda$ was $10^{-4}$, with $k = 2$, making it difficult to understand how LRN could have had much impact. Nonetheless it improved performance, though this was disputed by Simonyan & Zisserman (2015), but in our hands by less than DN (see Table 1). Ren et al. (2016) developed a unified mathematical framework for slightly modified batch, layer, and DN, combined it with an L1 regularizer, and showed that various forms of this (learned) regularized normalizer improved performance on CIFAR-10 and CIFAR-100 in a network with 3 convolutional and 2 fully connected layers, with the best performance by a modified batch norm. Their DN included in a unit's normalization pool all channels in a local spatial region about the unit. Giraldo & Schwartz (2019) explored a flexible, stimulus-dependent form of DN across space, based on the GSM, with learned parameters, that was added to the 2nd layer of a pretrained Alexnet to model contextual modulation in V1. Others have examined effects of DN on tasks in various biologically-motivated architectures (Coen-Cagli & Schwartz, 2013; Bertalmío et al., 2020). Burg et al. (2021) implemented a learnable form of DN in a model trained end-to-end to replicate spike counts of V1 neurons. The model included a single convolutional layer of 32 filters with batch normalization, followed by DN and a readout layer. The filters developed with no topology, that is, normalization weights were learned between each directed pair of channels.

The work most similar to our own was done independently by Pan et al. (2021). They considered a form of DN in which channels were partitioned into groups of 8, which normalized one another, followed by an affine transformation. They also considered adding a spatially local normalization pool restricted to a unit's own channel. For every unit, the affine transformations and the weights from every member of its normalization pool were learned. They also considered the DN Ren et al. (2016). They found that, compared to canonical normalizations, their channel normalization, but not the additional spatial normalization nor DN, improved performance on CIFAR-10 in shallow convolutional nets but not in deeper ones (4-5 or more layers). They attributed the improved performance on shallow networks to their channel normalization making activity distributions in early layers more Gaussian, Their channel normalization also showed some improvement over canonical normalizations for AlexNet on ImageNet. Our normalization pool size is learned (determined by the space constant of an exponential kernel), we examine pairings of divisive and canonical normalization which we find important to avoid failures to learn and improve performance relative to divisive alone, and we examine several properties – receptive fields and their Fourier power, manifold capacity, sparsity – that characterize ways in which the normalizations change representations.

## 3 METHODS

**Architecture.** We studied 8 models, each a variant of AlexNet (5 convolutional layers and 3 fully connected layers; Krizhevsky et al., 2012) with different normalization layers, with a Kaiming He initialization (He et al., 2015) and without pytorch local response normalization (LRN). The filters are 11x11 for ImageNet and 3x3 for CIFAR-100 in the first layer, 3x3 in all subsequent layers. The order of operations in each convolutional layer is ReLU, then DN if used, then canonical normalization if used (Divisive, Batch, Group, Layer).

**Normalization Formalisms.** For DN, the channels in a given layer develop topologically arranged on a line, while normalizing each other over a learnable distance specified by an exponential kernel. Thus the channels, while learning appropriate filters, are also learning appropriate linear arrangements that determine who normalizes whom. Given $n$ channels in a layer, numbered from 1 to $n$, we let $a_c(x)$ be the rectified output of the convolution with the filter of channel $c$ at 2D spatial position $x$. We take the unnormalized activation of this channel to be $a_c(x)^2$. We then divisively normalize, using as a "normalization pool" an exponentially weighted sum of the unnormalized activations of nearby channels at the same spatial position, to yield the unit's normalized activity $b_c(x)$:

$$b_c(x) = \frac{a_c(x)^2}{\left(k\left(1 + \frac{\alpha}{\lambda}\sum_{j=-4\lambda}^{4\lambda} a_{c+j}(x)^2 e^{-|j|/\lambda}\right)\right)^\beta} \quad (1)$$

Here, $\beta$, $\alpha$, $k$ and $\lambda$ are all learnable parameters, learned independently for each convolutional layer.

We also considered models in which each divisive normalization was followed by a "canonical" normalization: either batch, group, or layer. In all three, the normalization is of the form: $\tilde{z}_{n,j} = \gamma \frac{z_{n,j} - \mathbb{E}[z_n]}{\sqrt{Var[z_n]+\epsilon}} + \beta$. Here $\gamma$ and $\beta$ are learnable parameters. The subscript $n$ denotes the set that is normalized together. For example, in batch normalization, for an input of dimension $N \times c \times H \times W$, in which $c$ is the number of features, $H$ and $W$ the spatial dimensions and N the number of images in a batch. The mean and variance in this equation are calculated for a given feature across all of space and the whole batch. For layer normalization, the mean and variance are calculated across the spatial and feature dimensions for each image. For group normalization, the feature dimension is divided into 4 equal-sized groups in each layer, and the normalization is done within each group across space for each image.

**Hyperparameters.** Unless otherwise specified, the learning rate used in the models was .01. Batch sizes were 128. The initial normalization parameters were $\lambda = 10.$, $\alpha = .1$, $\beta = 1.$, $k = 10$, except for the Divisive model with no other normalizations, for which initial $\lambda = 1.$ and $k = 0.5$ to make learning reliable (further discussed in Results). The Weight initialization method followed that of He et al. (2015) in which weights are initialized with the same statistics for differing seeds. Specifically, the He formulation for ReLU activation functions is meant to keep the expected activation variances constant across layers. We used the same principle for our networks with ReLU plus DN and arrived at the same weight initialization. We then used the same initialization for combined divisive/canonical models. See Appendix A and B for more information.

The CIFAR training and validation images were resized to $32 \times 32 \times 3$ and horizontally flipped; Imagenet training images resized to $224 \times 224 \times 3$ and horizontally flipped; Imagenet validation images resized to $256 \times 256 \times 3$ and center cropped. Each color channel was always standardized.

## 4 RESULTS

**Top-1 Accuracy.** Tables 1-2 show top-1 accuracies at epoch 90 for the various models on ImageNet and CIFAR-100. The models are named by the type(s) of normalization used. For a name with two normalizations, the second normalization was applied immediately after the first, for example DivisiveBatch means divisive and then batch normalization were applied after each convolutional layer and ReLU. NoNorm indicates that none of the normalizations were used, simply a vanilla AlexNet. For ImageNet, these numbers are based on a single seed per model, as the final performance differed by no more than .1 - .2 points across seeds and it was not worth computational resources to do 30 seeds. The CIFAR-100 results show mean±stderr across 30 trials per model. Performance curves for CIFAR-100 (Fig. 1) show that performance differences were relatively stable from the earliest epochs.

For NoNorm and each canonical normalization, addition of DN almost always improved performance (exception: layer normalization for CIFAR-100). DivisiveBatch showed the best performance. Conversely, adding a canonical normalization almost always improved performance for DN (exception: Divisive and DivisiveLayer were statistically tied for CIFAR-100) and also improved reliability of learning. The Divisive model (without a canonical normalization) was less able to learn than the others; a parameter search on the DN parameters found that it learned reliably only for small values of $\lambda$ and $k$ (see Eq. 1). Applying a canonical normalization to standardize the activations after the DN

| Model | Validation Acc | Training Acc |
|---|---|---|
| NoNorm | 56.49 | 53.16 |
| NoNorm w LRN | 57.55 | 54.98 |
| Batch | 59.58 | 56.14 |
| Group | 58.57 | 57.35 |
| Layer | 59.87 | 58.32 |
| Divisive | 59.39 | 55.66 |
| DivisiveBatch | **61.33** | 59.27 |
| DivisiveGroup | 60.35 | **59.61** |
| DivisiveLayer | 60.1 | 58.32 |

Table 1: ImageNet Top-1 Accuracies

| Model | Validation Acc | Training Acc |
|---|---|---|
| NoNorm | $49.45 \pm .072$ | $56.00 \pm .04$ |
| Batch | $53.96 \pm .07$ | $65.79 \pm .04$ |
| Group | $51.55 \pm .07$ | $63.62 \pm .05$ |
| Layer | $52.06 \pm .07$ | $65.00 \pm .05$ |
| Divisive | $50.11 \pm .21$ | $58.46 \pm .32$ |
| DivisiveBatch | $\mathbf{54.88 \pm .07}$ | $\mathbf{66.35 \pm .03}$ |
| DivisiveGroup | $52.75 \pm .15$ | $63.92 \pm .05$ |
| DivisiveLayer | $50.18 \pm .11$ | $61.74 \pm .10$ |

Table 2: CIFAR-100 Top-1 Accuracies

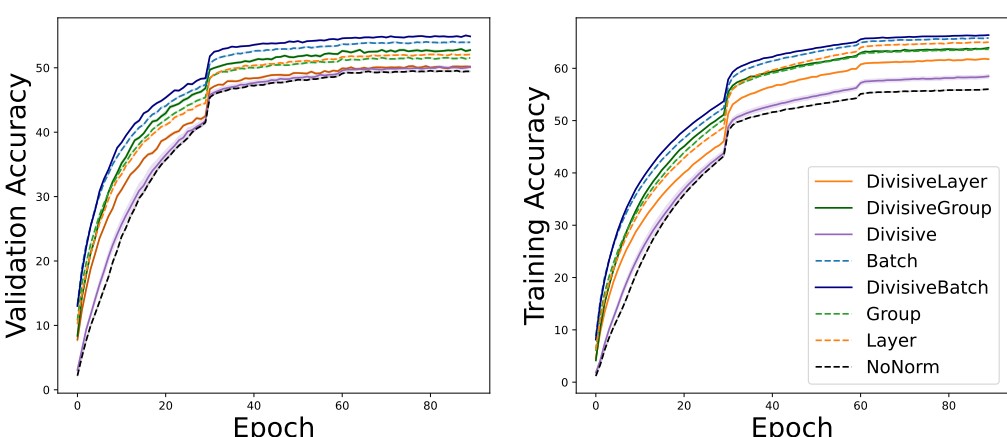

Figure 1: Performance curves for validation (left) and training (right) accuracy for each model across 30 trials on the CIFAR-100 dataset. Standard errors are plotted but are too small to see. Dashed curves are the models with a single type of normalization (canonical or divisive), black is the vanilla AlexNet (NoNorm), and solid curves are models with DN followed by a canonical normalization. Performance order is stable across epochs. Every 30 epochs, the learning rate is reduced by a factor of 10, hence the jumps in the curves.

made learning reliable. Additionally, across the DivisiveBatch, DivisiveGroup, and DivisiveLayer models, the divisive parameters quickly evolved to their final values, and were similar between the three different models for a given initial condition (Appendix C). While the models tended to perform better with a larger $\lambda$ and $k$ when we fixed these parameters (Appendix B), we also found that, when these parameters were learned, the initial divisive parameters chosen did not greatly change performance (Appendix C).

**Receptive Fields, Fourier Power, and Orientation Selectivity.** DN is a biologically motivated computation, seen throughout the visual cortical stream. Therefore, to better understand how it improves image classification performance, it is informative to study the receptive fields (RFs; i.e., filters) learned by the models. We focus on the 11x11 filters learned for each channel in the first layer of the ImageNet-trained models, which are the filters that are most interpretable and most easily compared to those seen in V1. The RFs of the models with DN show increased long-wavelength (Divisive and DivisiveLayer) or medium-wavelength (DivisiveBatch and DivisiveGroup) Fourier power compared to other models (Fig. 2). This can be seen in comparing the first 16 of the 96 RFs for the Divisive model (Fig. 3) to those of the NoNorm model (Fig. 4) (RFs for the other models are shown in the Appendix G). Compared to the Divisive model, the NoNorm model RFs have many more irregular and small scale structures, and fewer (and higher-frequency) grating-like RFs resembling the orientation-selective simple cells found in V1 (*e.g.*, Hubel & Wiesel, 1962). Consistent with these results, we found that addition of DN increased the median orientation selectivity of RFs in models with Batch or Group normalization (& no significant change in other models; Fig. 31, Appendix K).

**Feature Correlation.** Figs. 3-4 illustrate which filters are near one another (in the line topology along which features are arranged) and thus have developed while normalizing one another. In Fig. 28, Appendix J, we examine this in another way, plotting the layer-wise 1D correlations between feature

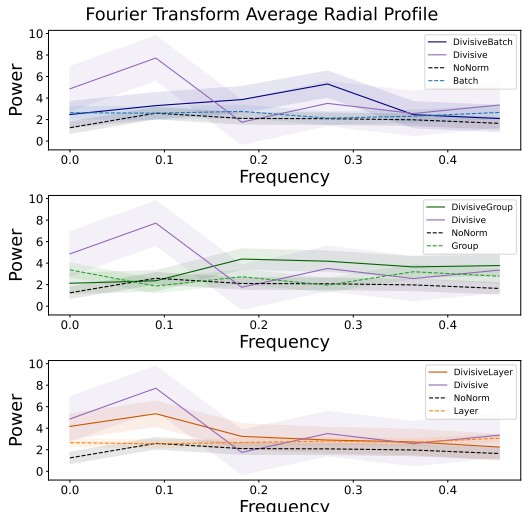

Figure 2: The radial Fourier power (*i.e.*, integrated over $\theta$ in polar coordinates) averaged over the 96 receptive fields (filters) in layer 1 of AlexNet for each architecture. Columns are red, green, and blue channels respectively. Rows show models with batch (blue lines), group (green lines) and layer (orange lines) respectively; darker color is divisive+canonical, lighter is canonical alone. The purple (Divisive) and black (NoNorm) lines are identical in all three rows of a given column, repeated for comparison with the various divisive+canonical and canonical models. Shading shows standard errors. While the small wavelength (high frequency) power is relatively similar from model to model, the long wavelength power is increased for the Divisive and DivisiveLayer models, while the mid-wavelength power is increased for the DivisiveGroup and the DivisiveBatch models.

weights as a function of feature separation. DN induces de- or anti-correlation of nearby features, and in some cases induces a reduction in correlation among all features regardless of separation.

**Replica-based Mean Field Manifold Geometry.** When an image classification model is trained, the activity patterns of N neurons for the images in a class form a manifold corresponding to that class in the N-dimensional space of activities. DiCarlo et al. (2012) argued that, in a feed-forward CNN, these object manifolds are hierarchically untangled such that, at higher layers, the individual classes are more linearly separable. A well performing model should produce object manifolds that are readily separable and therefore distinguishable. The system load of a model, $\alpha = \frac{P}{N}$, where P is the number of object manifolds and N is the number of neurons (Gardner, 1988; Chung et al., 2018) can measure the potential for a model to distinguish P manifolds. The classification capacity of a neural network is the maximum load for which most dichotomies of manifolds are linearly separable. Thus, a model that has higher performance on a specific dataset is expected to have higher classification capacity of object manifolds. This has been shown in recent studies (Stephenson et al., 2019; Chung S, 2020).

Using mean field theory methods developed by Chung et al. (2018), we measured the capacity of object manifolds and their geometric properties [1]. This depends on three quantities:

- **Radius**: The average size of a neural manifold across the classes considered.
- **Dimension**: The average embedding dimension of an individual object manifold from the perspective of a linear classification of manifold dichotomies.
- **Correlation**: The average correlation between the centers of the individual class neural manifold representations in state space. A lower correlation means that the individual neural manifolds would be more distinguishable (less entangled).

The manifold capacity – the ability to separate the object manifolds – is larger for smaller radius, dimension, and correlation.

In Figure 5, we plot the manifold capacity and properties for the NoNorm, Divisive, DivisiveBatch, and Batch models (results involving group and layer normalization, and changes in manifold parame-

---

[1]The analysis code used in this work is from `https://github.com/schung039/neural_manifolds_replicaMFT` from prior work by Stephenson et al. (2019).

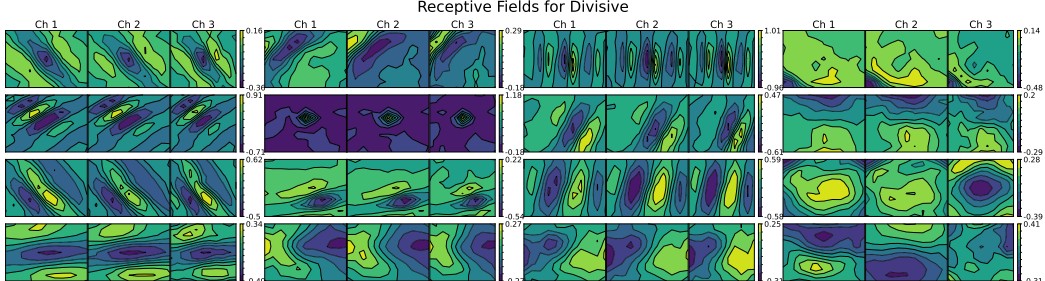

Figure 3: The receptive fields (RFs) for each of the 3 channels for the first 16 features along the topological line of 96 features in layer one for the Divisive model show many wide-set Gabor-like filters. (#'s 1-16 are arranged in English reading order, *i.e.* row 1 has #'s 1-4.)

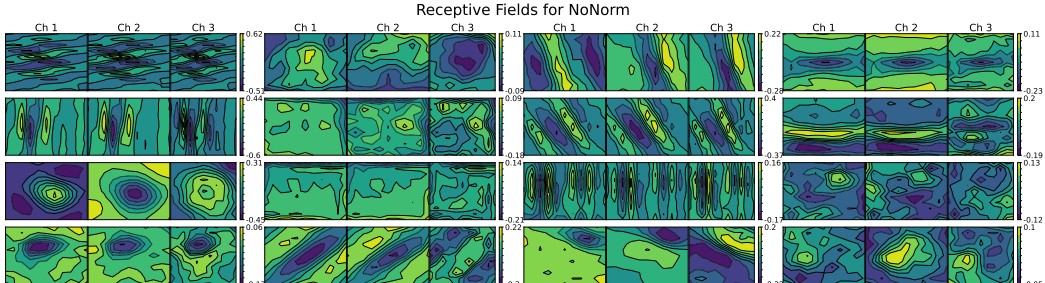

Figure 4: The receptive fields for each of the 3 channels for the first 16 features (as in Fig. 3) in layer one for the NoNorm model show a wider variety of filters with more irregularity and more small-scale features. In Figs. 3-4: the 16 appear fully representative, space precludes showing all 96.

ters after each individual normalization or convolutional layer, are presented in Appendix D). In the early layers, models using batch and/or divisive normalization, and particularly the Batch model, tend to have lower capacities and higher manifold dimension, radii, and correlation compared to NoNorm. The most notable change is the pronounced jump in correlation between manifold centers at interior ReLU layers. By the fully connected layers, the situation has reversed, with the two models involving DN having largest manifold capacity and smallest radius and dimensions, followed by the Batch model, with NoNorm last (for correlation, the three normalized models have the same ordering, but NoNorm is smallest). It is intriguing that somehow, by creating early representations with lower capacity and higher-dimensional object manifolds, the network with these normalizations can arrive in the end at a higher-capacity representation with lower-dimensional object manifolds.

**Sparsity.** Firing in sensory cortex is relatively sparse, with long-tailed activity distributions (Shafi et al., 2007; Hromadka et al., 2008; O'Connor et al., 2010; Barth & Poulet, 2012), particularly for responses to natural stimuli (Froudarakis et al., 2014). Both higher areas of visual cortex and higher layers of AlexNet and other deep nets trained for visual object recognition respond better to stimuli with naturalistic statistics than those without them (Zhuang et al., 2017), and in the deep net higher layers this property is strongly correlated with sparsity of firing and not with any other tested property (Zhuang et al., 2017). Sparseness of neural firing has been postulated to have many functions, including improving discrimination, learning, memory capacity, disentangling and linear readout, and other properties by reducing overlap between activity patterns for different stimuli (*e.g.*, Olshausen & Field, 2004; Ganguli & Sompolinsky, 2012). More generally, by various criteria there is an optimal level of sparsity, such as optimizing a tradeoff between discrimination and generalization (Barak et al., 2013) or optimizing the dimensionality of a representation (Litwin-Kumar et al., 2017).

While there are various measures of sparsity, we use the Gini index, which is designed to capture the kind of sparsity represented by heavy-tailed distributions. The Gini index is a general metric for measuring inequality, typically across countries. It is defined in terms of the Lorenz curve: a graph of the cumulative activations in each layer (or income) as a function of the cumulative % of activations (or incomes) counted, where the activations are arranged from smallest to largest. For the most extreme inequality, the curve would be at zero as the x-axis moves from 0 to 100%, and then jump to 1 at 100%, meaning all of the activation is in a single unit. In contrast, the diagonal

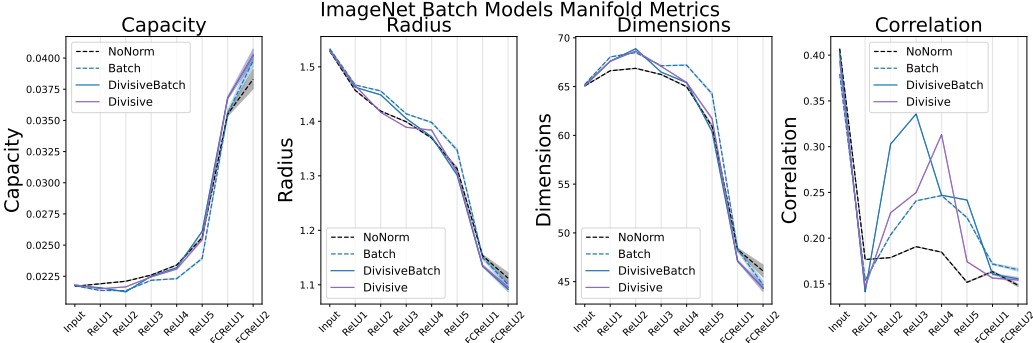

Figure 5: The mean field theory manifold capacity is calculated on the 50 least correlated classes (Cohen et al., 2020) for the ImageNet training data, using responses to 100 images randomly selected for each class. Results are shown for the ReLU layers of the NoNorm, DivisiveBatch, Batch, and Divisive models. Vertical grey lines indicate ReLU layers. Error bars denote the standard error across five different seeds of these 100 randomly selected images in each class.

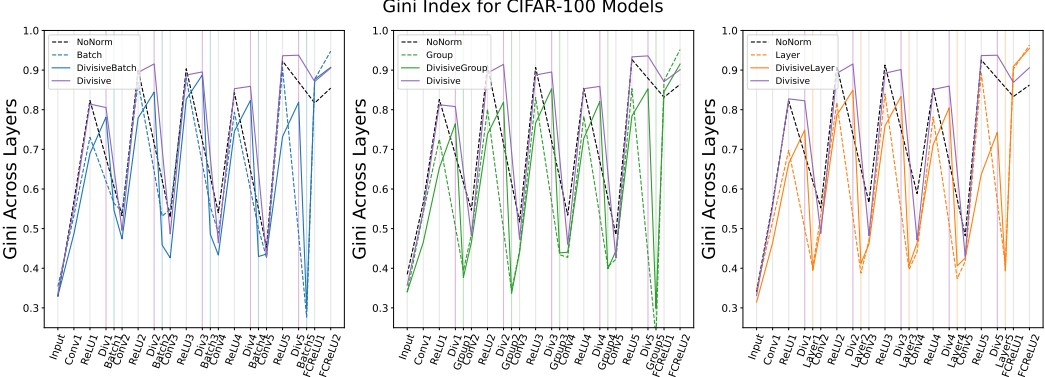

Figure 6: The sparsity, as measured by the Gini index ($1 = $ most sparse, $0 = $ all activations equal), for the validation set of images at epoch 90, for CIFAR-100 models. Results are show for canonical (*e.g.*, Batch; dashed color) and combined divisive-canonical (*e.g.*, Divisive Batch, solid color) models involving the batch norm (blue, left panel), the group norm (green, middle), and the layer norm (orange, right), as well as NoNorm (black) and Divisive (purple) models (repeated in all 3 panels, for comparisons). The vertical purple lines indicate where the divisive normalization occurs for models that have it. Similarly the vertical blue, green, and orange lines indicate where a canonical normalization, *e.g.* batch, group, or layer norm respectively, occurs.

represents equality of all activations. The Gini index is defined as the area between the diagonal and the Lorenz curve, divided by the area under the diagonal. Thus, it is 1 for the case of most extreme inequality or sparseness, and 0 for complete equality.

We examined the Gini index in each layer for the models trained on CIFAR-100. (Distributions of probabilities of activation are in Appendix F.) The Gini index is slightly increased (*i.e.*, sparsity is increased) by a divisive normalization layer, while canonical normalization layers strongly reduce the Gini index. In the first four layers, after all normalizations, the Divisive and NoNorm models are most sparse, while canonical and combined divisive-canonical models are less so. By the final, fully-connected layers, the canonical models are most sparse (mean probability of activation $\sim 10\%$, Figs. 17-19, Appendix F), followed by combined divisive-canonical and Divisive (probability $\sim 20\%$), and NoNorm model (probability $\sim 30\%$), with the exception that for layer norm, the DivisiveLayer model is equally as sparse as the Layer model. Thus divisive normalization strongly impacts sparsity of activations, generally fixing sparsity of the final representation at an intermediate level ($\sim 20\%$) but producing stronger sparsity when combined with layer norm.

**Out of Distribution (OOD) Images.** We investigated whether DN might improve OOD performance in two ways. First, we considered a black box adversarial attack of adding Gaussian noise to images

(Fig. 26, Appendix H). DivisiveGroup performed best against moderate-strength attacks, while DivisiveBatch and Divisive Layer performed best against the strongest attacks we tried, providing preliminary evidence that DN can improve OOD performance. Second, given the bias of CNNs to recognize texture while humans are biased toward shape, we considered images in which the texture of one category was combined with the shape of another (Geirhos et al., 2018) (Fig. 27, Appendix I). Among images for which the network chose either the texture or the shape category, DN led to an increase in the % of shape choices, but the networks with DN also chose other categories more often. This gives some hope of DN improving shape sensitivity, but does not allow clear conclusions.

## 5 DISCUSSION

Divisive normalization is a phenomenological model that captures nonlinear response properties seen ubiquitously in sensory cortical areas (Carandini & Heeger, 2012). While many functions have been proposed for these properties, why cortical neurons have them remains an open question. This paper has been a first exploration of what a learnable DN may achieve computationally in the context of AlexNet and canonical CNN normalizations. We examined how it changed both performance and the learned representations in an object recognition task.

Channels were learned in a line topology, with the normalization pool of each unit determined by the channels to either side of it weighted by a learnable exponential kernel. DN improved performance on ImageNet and CIFAR-100 by a few percent, but it was prone to failing to learn. We therefore studied combined models, in which DN was followed by batch, group, or layer norm, which learned much more reliably. A similar stabilizing role biologically might be played by forms of homeostatic plasticity, which keep neural activities in a desired operating range (Turrigiano, 2011). (It could be interesting to substitute homeostatic plasticity for canonical normalizations in future studies (Shen et al., 2021)). For batch, group, and no normalization, addition of DN always improved performance, with the DivisiveBatch model performing best of all studied models.

As measures of how representations are changed by the normalizations, we studied the receptive fields (RFs) of the models' first layer, calculated the sparsity in each layer, and analyzed the neural manifold geometry in each layer. The Divisive model's first layer RFs have increased power at large wavelengths and the combined models similarly have increased power at large and/or mid-scale wavelengths. For DivisiveBatch and DivisiveGroup, the RFs have increased orientation selectivity. Under DN, nearby RFs, which can normalize one another, develop to be de- or anti-correlated relative to other RFs, and in some cases all RFs are de-correlated under DN compared to RFs in models without DN. We found that DN can improve performance on OOD images formed by adding Gaussian noise, and by one measure (but not another) can increase bias for recognizing objects by their shape rather than their texture. Models with DN generally led to a sparsity of final layer activations intermediate between those of purely "canonical" normalizations (more sparse) and unnormalized models (less sparse). Surprisingly, in earlier layers, the normalized models (except purely divisive) are less sparse than unnormalized ones, which somehow leads to a more sparse final representation. A similar reversal is seen in manifold capacity and its associated geometric quantities. Normalized models produce representations in earlier layers with lower manifold capacity, yet produce increased capacity in the final layer, with DN models having the highest capacity.

This study raises many questions. Why does DN produce the observed representational changes, and are these correlates of improved performance also causal? How is it that a decrease in sparsity or manifold capacity in intermediate layers can yield an increase in the final layer? Under what conditions is it best for DN to come from more dissimilar rather than more similar RFs? Might DN have advantages that would be better revealed by other tasks, such as object recognition in more cluttered scenes, object segmentation, or analysis of videos? How will divisively normalizing across space as well as channels, and considering 2- or 3-D topologies, alter results?

Here, we have explored how DN alters computations and representations in a simple context, learning object recognition in a CNN, possibly along with other canonical CNN normalizations. Ultimately we would like to understand visual systems that replicate both neural (*e.g.*, Schrimpf et al., 2018) and psychophysical behaviors of biological systems. This will require learning with more complex and richer contexts, constraints, and combinations of mechanisms including DN, which may alter the function of DN. Here we begin by learning how DN functions in simpler systems that can learn visual tasks. we hope this contributes to a fruitful interaction between studies of deep networks and of neuroscience that advances both.

## REPRODUCIBILITY

All software used in this project will be deposited in a publicly accessible github repository no later than the time of the 2022 ICLR meeting.

## ACKNOWLEDGEMENTS

Funded by NSF grants IIS-1704938 and DBI-1707398 and the Gatsby Charitable Foundation.

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

CONTENTS

**Part**

# Appendix

## Table of Contents

## A   INITIALIZATION OF WEIGHTS AND NORMALIZATION PARAMETERS

We utilize a Kaiming He initialization (He et al., 2015). The traditional Kaiming He initialization is formulated for ReLU activation functions by calculating the variance of the output for a layer and then obtaining values for the initial weights such that the network will remain stable. In the original formulation, He utilizes the variance of the response in each layer:

$$\mathbf{y}_l = W_l\mathbf{x}_l + \mathbf{b}_l$$

Similar to the original He paper, we take $\mathbf{b}_l$ to be 0 for all $b'_l s$. Additionally, $w_l$ and $x_l$ are independent of one another and the $y_l, w_l$ and $x_l$ variables are random variables.

Then they pass $\mathbf{y}_l$ through a ReLU function denoted as:

$$\mathbf{x}_l = f(\mathbf{y}_{l-1})$$

where in the initial He formulation, $f(\cdot)$ is the rectifying function, $max(0, \mathbf{y}_{l-1})$. In our formulation, instead of a ReLU, the function goes through the divisive normalization after the ReLU.

$$\mathbf{x}_l = f(\mathbf{y}_{l-1}) = \frac{max(0, \mathbf{y}_{l-1})^2_c}{\left(k(1 + \frac{\alpha}{\lambda}\sum_{j=1}^{N} max(0, \mathbf{y}_{l-1})^2_j e^{-|c-j|/\lambda})\right)^{\beta}}$$

The original He derivation does the following: It takes the variance of $y_l$, taking $y_l$ and $w_l$ to be mean zero.

$$Var[y_l] = n_l Var[w_l x_l]$$
$$Var[y_l] = n_l(\mathbb{E}[w_l^2 x_l^2] - \mathbb{E}[w_l x_l])^2)$$
$$Var[y_l] = n_l(\mathbb{E}[w_l^2]\mathbb{E}[x_l^2] - \mathbb{E}[w_l x_l])^2)$$

Here we can rewrite and substitute $\mathbb{E}[x_l^2] = Var[x_l] + \mathbb{E}[x_l]^2$ and $\mathbb{E}[w_l^2] = Var[w_l] + \mathbb{E}[w_l]^2$

$$Var[y_l] = n_l((Var[x_l] + \mathbb{E}[x_l]^2)(Var[w_l] + \mathbb{E}[w_l]^2) - \mathbb{E}[w_l x_l]^2)$$
$$Var[y_l] = n_l(Var[x_l]Var[w_l] + Var[w_l]\mathbb{E}[x_l]^2 + Var[x_l]\mathbb{E}[w_l]^2 + \mathbb{E}[x_l]^2\mathbb{E}[w_l]^2 - \mathbb{E}[w_l x_l]^2)$$
$$Var[y_l] = n_l(Var[w_l]Var[x_l] + Var[w_l]\mathbb{E}[x_l]^2 + Var[x_l]\mathbb{E}[w_l]^2)$$
$$Var[y_l] = n_l(Var[w_l](\mathbb{E}[x_l^2] - \mathbb{E}[x_l]^2) + Var[w_l]\mathbb{E}[x_l]^2)$$
$$Var[y_l] = n_l(Var[w_l](\mathbb{E}[x_l^2]))$$

Above we used the independence of the variables $x_l$ and $w_l$ to say $\mathbb{E}[w_l x_l]^2 = \mathbb{E}[w_l]^2\mathbb{E}[w_l]^2$. Additionally, $\mathbb{E}[w_l] = 0$ as the weights are initialized with mean 0. In the initial He form, $x_l = max(0, y_{l-1})$, which as a ReLU does not have mean 0, but $y_{l-1}$ does have mean zero. Using symmetry of the ReLU function, we can see that $\mathbb{E}[x_l] = \frac{1}{2}Var[y_{l-1}]$. :

By still modifying the variance of $y_l$ to consider our the added stages of going through the divisive normalization function

$$Var[y_l] = n_l Var[w_l x_l]$$
$$Var[y_l] = n_l Var[w_l]Var[x_l]$$
$$Var[y_l] = n_l Var[w_l]Var[\frac{max(0, \mathbf{y}_{l-1})^2_c}{k^{\beta}\left(1 + \frac{\alpha}{\lambda}\sum_{j=1}^{N} max(0, \mathbf{y}_{l-1})^2_j e^{-|c-j|/\lambda}\right)^{\beta}}]$$

The purpose of this formulation is ultimately to get an expression for $Var[w_l]$ such that the weights will not blow up or die out across subsequent layers. Thus, we need to hone in on and simplify the

variance of the quotient. Recalling that the variance of a quotient of two random variables x and y is in the form $Var[x/y] = Var[x] + Var[y] - 2Var[x]Var[y]corr(x, y)$ such that

$$Var[\frac{max(0, \mathbf{y}_{l-1})_c^2}{k^\beta \left(1 + \frac{\alpha}{\lambda} \sum_{j=1}^N max(0, \mathbf{y}_{l-1})_j^2 e^{-|c-j|/\lambda}\right)^\beta}] =$$

$$= Var[max(0, \mathbf{y}_{l-1})_c^2] + Var[k^\beta \left(1 + \frac{\alpha}{\lambda} \sum_{j=1}^N max(0, \mathbf{y}_{l-1})_j^2 e^{-|c-j|/\lambda}\right)^\beta] -$$

$$2Var[max(0, \mathbf{y}_{l-1})_c^2] \times Var[k^\beta \left(1 + \frac{\alpha}{\lambda} \sum_{j=1}^N max(0, \mathbf{y}_{l-1})_j^2 e^{-|c-j|/\lambda}\right)^\beta] \times corr(\cdot, \cdot)$$

For our purposes, we will assume the correlation between the numerator and the denominator is small enough to be negligible. (This is still possibly a big assumption because the denominator does have a "self" term, but we are assuming this self term is small relative to the sum of all of the other terms to make the math tenable. This makes a little more sense when we assume large $\lambda$.

$$Var[\frac{max(0, \mathbf{y}_{l-1})_c^2}{k^\beta \left(1 + \frac{\alpha}{\lambda} \sum_{j=1}^N max(0, \mathbf{y}_{l-1})_j^2 e^{-|c-j|/\lambda}\right)^\beta}] =$$

$$= Var[max(0, \mathbf{y}_{l-1})_c^2] + Var[k^\beta \left(1 + \frac{\alpha}{\lambda} \sum_{j=1}^N max(0, \mathbf{y}_{l-1})_j^2 e^{-|c-j|/\lambda}\right)^\beta]$$

$$= \frac{1}{2} Var[(y_{l-1})_c^2] + Var[k^\beta \left(1 + \frac{\alpha}{\lambda} \sum_{j=1}^N max(0, \mathbf{y}_{l-1})_j^2 e^{-|c-j|/\lambda}\right)^\beta]$$

Trying to actually take the variance of the second term when $\beta, \lambda, \alpha$ and $k$ are all variable parameters, the math becomes intractable. Particularly because any given power of value $\beta$ will give an intractable binomial expansion. Since this calculation only depends on the initial values of the parameters, we initialize $\beta = 1$.

$$= \frac{1}{2} Var[(y_{l-1})_c^2] + Var[k \left(1 + \frac{\alpha}{\lambda} \sum_{j=1}^N max(0, \mathbf{y}_{l-1})_j^2 e^{-|c-j|/\lambda}\right)]$$

Now in order to make the rest of this tractable, we hone in on the second term. The term is calculating the variance of the convolution of the ReLU output with an exponential kernel along the feature dimension.

If we notice that $e^{-|c-j|/\lambda}$ will always be bounded by 1, then we can see there will be an upper bound for the value within the parentheses when considering some maximum activation value for a given layer. The input images have dimensions $H \times W \times N$, where H is the height, W is the width and N. Further, for an exponential probability distribution, 98% of the distribution falls within $4\lambda$ of the center. Therefore, it is a reasonable assumption to approximate an upper bound of the variance to be in terms of $4\lambda$ neighbors.

$$Var[k \left(1 + \frac{\alpha}{\lambda} \sum_{j=1}^N max(0, \mathbf{y}_{l-1})_j^2 e^{-|c-j|/\lambda}\right)] \leq Var[k \left(1 + \frac{\alpha}{\lambda} 4\lambda max(0, \mathbf{y}_{l-1})_{max}^2\right)]$$

$$\leq Var[k \left(1 + 4\alpha max(0, \mathbf{y}_{l-1})_{max}^2\right)]$$

$$\leq Var[k \left(1 + \frac{\alpha}{\lambda} N max(0, \mathbf{y}_{l-1})_{max}^2\right)]$$

This upper bound corresponds to using a uniform kernel rather than an exponential kernel in which each of the neighbors would be considered equally important up to N neighbors or more realistically,

$4\lambda$ neighbors. If the quantity $4\alpha$ is sufficiently small then this term will drop out when calculating the variance. This bounds the variance of our expression by zero, and since the variance can only be a nonzero term, it is approximately zero. Further noting that $Var[1 + x] = Var[x]$ and $Var[kx] = k^2 Var[x]$, where k is a constant, we can rewrite the variance

$$Var[k\left(1 + \frac{\alpha}{\lambda}\sum_{j=1}^{N} max(0, \mathbf{y}_{l-1})_j^2 e^{-|c-j|/\lambda}\right)] \leq k^2 Var[(1 + 4\alpha max(0, \mathbf{y}_{l-1})_{max}^2)]$$

$$\leq k^2 Var[(4\alpha max(0, \mathbf{y}_{l-1})_{max}^2)]$$
$$\leq 16k^2\alpha^2 Var[(max(0, \mathbf{y}_{l-1})_{max}^2)]$$
$$\leq 16k^2\alpha^2 \frac{1}{2} Var[(\mathbf{y}_{l-1}{}_{max}^2)]$$
$$\leq 8k^2\alpha^2 Var[(\mathbf{y}_{l-1}{}_{max}^2)]$$

Thus we can rewrite an approximate upper bound for the variance of the divisive normalization term below:

$$Var[\frac{max(0, \mathbf{y}_{l-1})_c^2}{k^\beta \left(1 + \frac{\alpha}{\lambda}\sum_{j=1}^{N} max(0, \mathbf{y}_{l-1})_j^2 e^{-|c-j|/\lambda}\right)^\beta}] \approx \frac{1}{2}Var[(y_{l-1})_c^2] + Var[k\left(1 + \frac{\alpha}{\lambda}\sum_{j=1}^{N} max(0, \mathbf{y}_{l-1})_j^2 e^{-|c-j|/\lambda}\right)]$$

$$\leq \frac{1}{2}Var[(y_{l-1})_c^2] + 8k^2\alpha^2 Var[(\mathbf{y}_{l-1}{}_{max}^2)]$$

Empirically, we see early in training that the activations after the ReLUs layers for the NoNorm, Divisive, DivisiveBatch and the Batch models are all on the order of 1. Thus, it is reasonable to assume that if we use a small enough $\alpha$ and $k$ upon initialization, then the second term will become negligible relative to the first term so we can approximate the overall $Var[y_l]$:

$$Var[y_l] \approx n_l Var[w_l]\frac{1}{2}Var[(y_{l-1})_c^2]$$

$$Var[y_L] \approx Var[(y_1)_c^2]\left(\Pi_{l=2}^{L}\frac{1}{2}n_l Var[w_l]\right)$$

Using similar arguments made in the Kaiming He initialization, we want to avoid the magnitudes of the input signals from growing unboundedly. Thus, we can approximately set a sufficient condition for stable learning using the Kaiming He initialization bet requiring:

$$\frac{1}{2}n_l Var[w_l] = 1, \forall l$$

This allows us to initialize the weights of the Divisive model to be zero mean with a standard deviation of $\sigma = \sqrt{\frac{2}{n_l}}$

Through this section of the appendix, we have found a set of values for the Divisive normalization parameter such that we can use Kaiming He initialization for the weights when implementing Divisive normalization after each ReLU in AlexNet use the Kaiming He. If we initialize $\beta$=1 and have $\alpha$ and $k$ to be reasonably small, we can address the vanishing/exploding gradient problem. Namely, these values for $\alpha$ and $k$ are .1 and .5 respectively. However, despite addressing vanishing/exploding gradient problem, we found the Divisive model does not reliably learn in isolation when placed in all five layers. This helped motivate including the more canonical normalizations such as Batch norm, Group norm and Layer norm after the Divisive normalization layers. The next section disucsses our parameter exploration.

# B    DIVISIVE NORMALIZATION PARAMETER CHOICES

In the prior section we determined a stable initialization for the divisive parameters to avoid the exploding/vanishing gradient problem by leaning on the Kaiming He initialization. While this derivation required we fix $\beta$ to be 1.0 and keep $\alpha$ and $k$ small, it left $\lambda$ free to vary at initialization. Because of this, we explored the parameter space for the Divisive normalization on the CIFAR-100 dataset to determine whether the initial values of these parameters impact learning performance. In particular, we considered initial values for $\lambda$ of 1, 2, 5, and 10 and for $k$ of .1, .5, 1 and 10. Unfortunately, the Divisive model failed to learn reliably when varying these two parameters, so for the Divisive model we used initial values of 1.0 for $\lambda$ and 0.5 for $k$. Varying values beyond these usually did not produce Divisive models that could learn. However, the combined models, in which Divisive normalization was paired with Batch, Layer, or Group normalization, learned for a wide range of initial values. Figures 7, 8, 9 show the validation accuracies of these models for each different parameter pairing of $\lambda$ and $k$. Typically, for the DivisiveBatch model, the initial values did not drastically impact performance. However, for the DivisiveGroup model, there seemed to be improved performance for larger $k$ and $\lambda$ values. Further, the DivisiveLayer model performed a few points better when the initial $\lambda$ values were larger. The initial k value did not seem to have much of an impact on the DivisiveLayer model. Because of this, all models other than Divisive in the main paper have the same initial values. We chose to initialize k to be 10 and $\lambda$ to be 10 both on the CIFAR-100 dataset and on ImageNet.

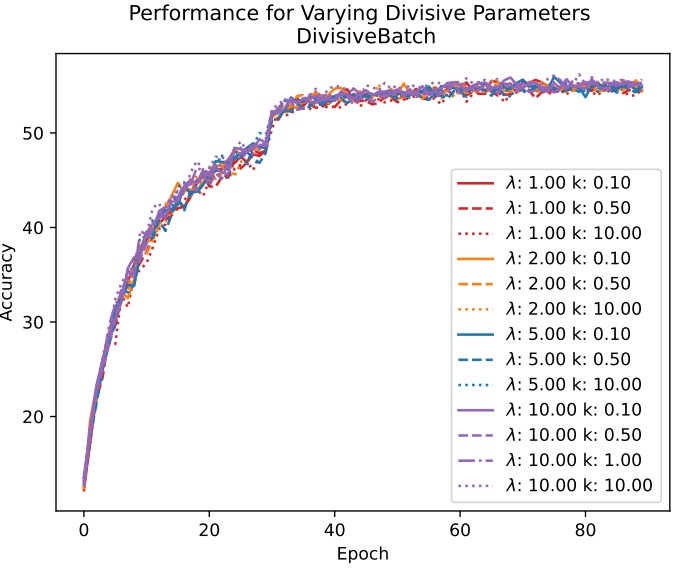

Figure 7: Dependence of performance on initial normalization parameters $\lambda$ and $k$ for the Divisive-Batch model. Red, orange, blue, and purple indicate initial $\lambda$ values of 1, 2, 5, and 10 respectively. The solid, dashed, dashdot, and dotted lines correspond to $k$ values of .1, .5, 1 and 10 respectively. The initial value of $\lambda$ does not obviously impact performance. There appears to be very modest improvement with larger $k$ values.

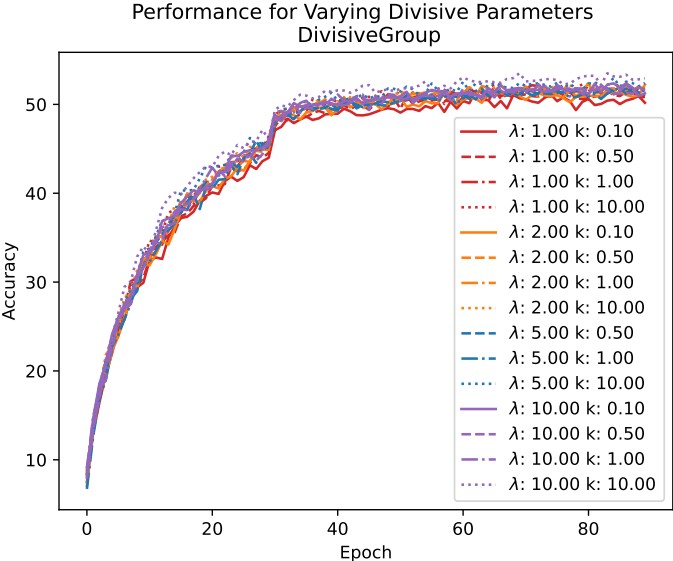

Figure 8: Dependence of performance on initial normalization parameters $\lambda$ and $k$ for the DivisiveGroup model. Larger $\lambda$ values result in a modest performance improvement, as does larger $k$.

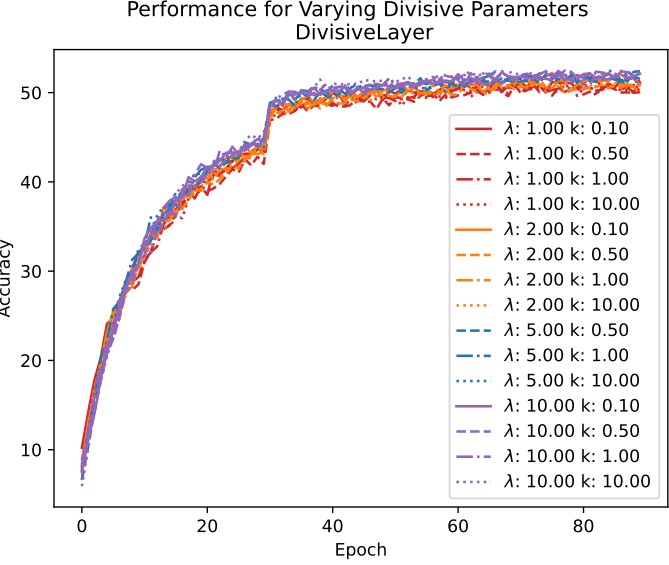

Figure 9: Dependence of performance on normalization parameters $\lambda$ and $k$ for the DivisiveLayer model. There appears to be a clear preference in the DivisiveLayer model to initialize with a larger $\lambda$ and $k$. This improvement is more pronounced relative to the DivisiveGroup model in which a larger $\lambda$ only gave modest improvement

## C    DIVISIVE NORMALIZATION PARAMETERS LEARNED

Using the parameter initializations explored in the prior section we plot the resulting normalization parameters that develop. All of the models learn slightly different divisive normalization parameters depending on the initial conditions. The Divisive model did not learn reliably enough for such a parameter search to be informative. However, there are some common trends across the combined models. For example, generally $\lambda$ and $k$ are relatively stable throughout learning. In contrast, the $\alpha$ and $\beta$ values change a lot initially and settle to some steady state after about 20 epochs.

In the DivisiveBatch model, the learned $\alpha$ value tends to get larger with a larger initial $\lambda$ and $k$. Such a trend is less clear in the DivisiveLayer model. In the DivisiveGroup model, it is less clear. For example, in layers 3 and 5, it appears a larger $\lambda$ will give a smaller $\alpha$.

Across all of the models in all of the layers, the $\beta$ values range from 0 to 2. In the DivisiveBatch and DivisiveGroup models, they tend to be around 1 or less except in the last layer where $\beta$ gets much larger. In the DivisiveLayer model, the learned $\beta$ parameter steadily increases across the layers. Meanwhile, the learned $\alpha$ steadily decreases across the layers and then increases again in layer 5. We see similar behavior in the DivisiveBatch model, in which $\alpha$ tends to get smaller regardless of the other parameter initializations but then begins to increase again in layers 4 and 5. For the DivisiveGroup model, the $\alpha$ values tend to be between 0 and 1 throughout all five layers.

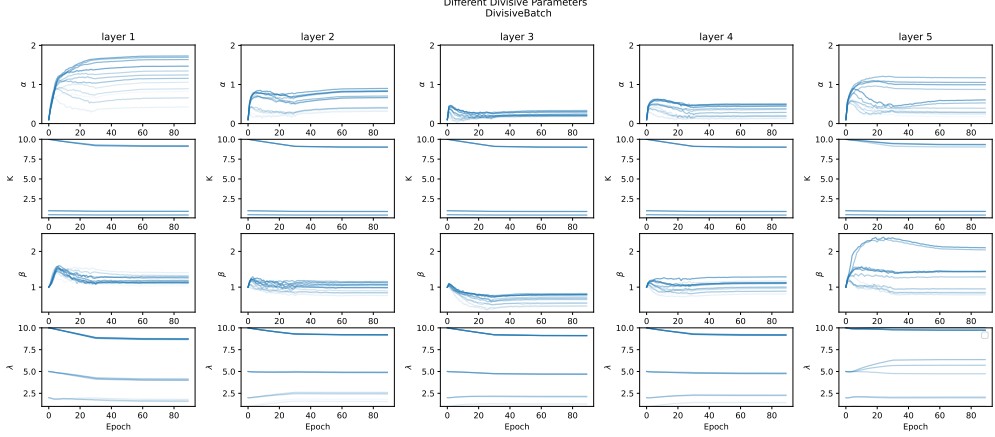

Figure 10: The different normalization parameters learned for the DivisiveBatch model over epochs. Each curve represents a slightly different set of initial conditions. Darker lines correspond to a larger initial $\lambda$. It is clear that the k values and $\lambda$ values do not vary much.

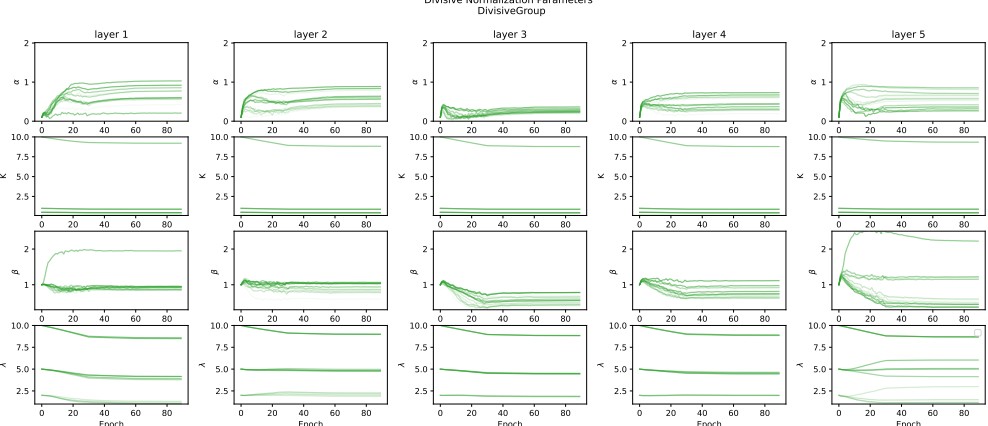

Figure 11: The different normalization parameters learned for the DivisiveBatch model over epochs. Each curve represents a slightly different set of initial conditions. Similar to the DivisiveBatch model, the parameters roughly approach a steady state after about 20 epochs.

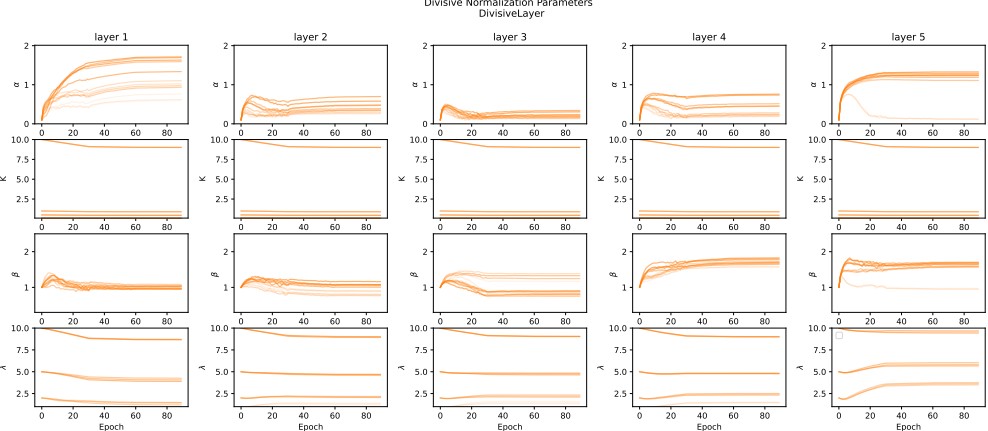

Figure 12: The different normalization parameters learned for the DivisiveLayer model over epochs. Each curve represents a slightly different set of initial conditions. In layer 3, $\beta$ is larger for smaller $\lambda$ and $k$, but in layer 2 and 5, it modestly gets smaller.

# D  NEURAL MANIFOLD ANALYSIS WITH GROUP AND LAYER NORM MODELS

While we only discussed a few of the normalized models for the Replica-based mean field geometry analysis in section 4 (primarily Batch, Divisive, and DivisiveBatch models), the other normalized models exhibit similar behavior. We also include the same results as Figure 5, but showing results after each normalization and convolutional layer rather than only for the ReLU layers. In the combined models, at a given layer, the divisive normalization tends to decrease the capacity or leave it unchanged.

The correlations between the manifolds are much more complex when looking at all of the layers. When only looking at the ReLUs, the normalized models have much higher correlations at the interior layers. However, when considering the convolutional and normalization layers, the normalized models have lower correlations. The convolutional layers increase the correlations between the manifolds, while ReLUs decrease the correlations layer-wise. For normalized models, after going through a convolutional-ReLU pair, the variation in the correlations is still more narrow than the variation in the NoNorm model. In other words, the correlation after a convolutional layer is higher in a NoNorm model relative to a normalized model, but the correlation after the ReLU is lower relative to that in a normalized model.

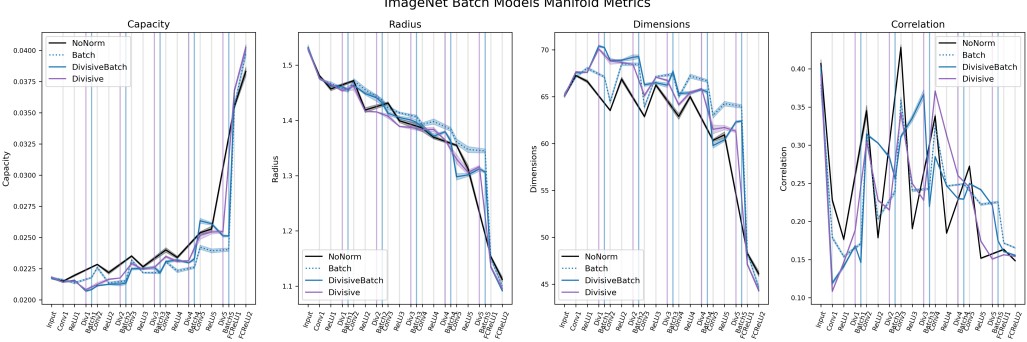

Figure 13: The mean field theory manifold capacity is calculated on the 50 least correlated classes (Cohen et al., 2020), using responses to 100 images randomly selected for each class, for the ReLU layers of the NoNorm, DivisiveBatch, Batch, and Divisive models. Vertical purple lines indicate where divisive normalization would occur, vertical blue lines indicate where Batch normalization would occur, and vertical grey lines indicate ReLU or convolutional layers. Error bars denote the standard error across five different samples of these 100 randomly selected images in each class. For the models with Batch normalization, there is a consistent jump in number of dimensions that describe the neural manifolds in the early layers. There is also a reduction in capacity in the early layers, more prominently in the Batch model. After each divisive normalization layer in the Divisive model, the correlation is consistently reduced. In the DivisiveBatch model, the divisive normalization layer decreases the correlation in layers 2, 4, and 5. Batch normalization consistently decreases the correlation between the neural manifolds in each layer, while in the Batch model, it consistently increases the correlation.

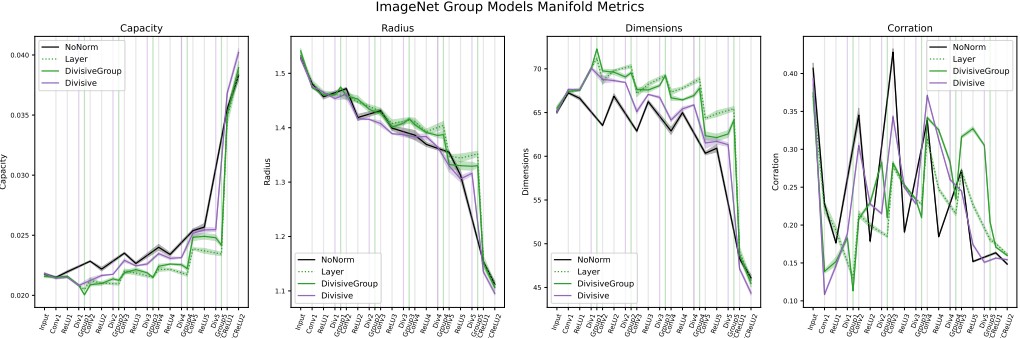

Figure 14: The mean field theory manifold capacity is calculated on the 50 least correlated classes Cohen et al. (2020) with 100 images randomly selected for each class for the ReLU layers of the NoNorm, DivisiveGroup, Group, and Divisive models. Vertical purple lines indicate where divisive normalization would be present, vertical green lines indicate where Group normalization would be present, and vertical grey lines indicate ReLU or convolutional layers. Error bars denote the standard error across five different samples of these 100 randomly selected images in each class.

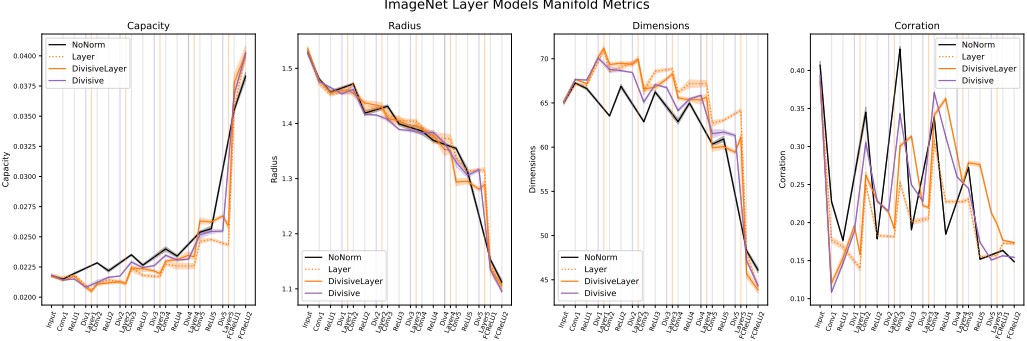

Figure 15: The mean field theory manifold capacity is calculated on the 50 least correlated classes Cohen et al. (2020) with 100 images randomly selected for each class for the ReLU layers of the NoNorm, DivisiveLayer, Layer, and Divisive models. Vertical purple lines indicate where divisive normalization would be present, vertical orange lines indicate where Layer normalization would be present, and vertical grey lines indicate ReLU or convolutional layers. Error bars denote the standard error across five different samples of these 100 randomly selected images in each class.

# E    COMPARISON BETWEEN THEORETICAL AND EMPIRICAL MANIFOLD CAPACITY

The manifold capacity reported in section 4 is that calculated using a replica-based mean field theory. The manifold capacity can be also measured empirically with a bisection search to find a critical number of features such that the fraction of linearly separably manifold dichotomies is close to 1/2 Chung et al. (2018); Cohen et al. (2020). As a sanity check, Figure 16 plots the mean field theory capacity calculations relative to the empirically calculated manifold capacity.

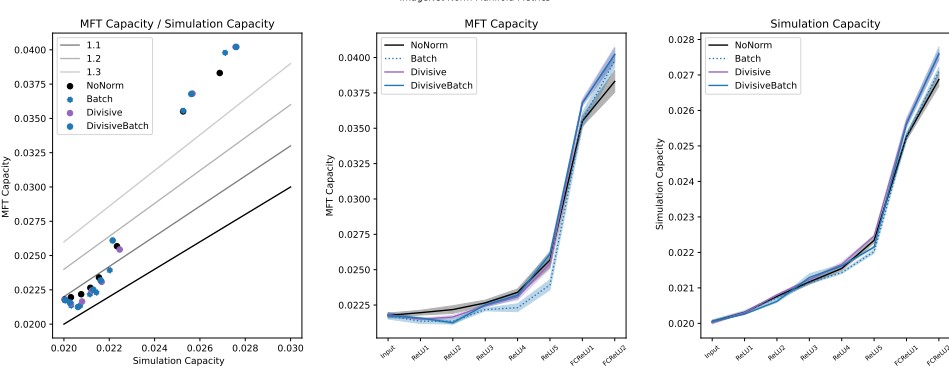

Figure 16: Left shows the Mean Field Theory (MFT) capacity plotted relative to the simulation capacity values layer by layer. The black line is the unity line. The second plot shows the layer-wise MFT capacities with the standard error plotted for each ReLU in the network. The third plot shows the layer-wise simulation capacities plotted for each ReLU. Deviations from the simulation capacity in the MFT capacity plot could indicate real changes in the underlying neural manifold geometry.

## F    SPARSITY OF ACTIVATION

In the main text we presented the Gini index as a measure of sparsity of the distribution of activations. Here we present the full activity distributions across layers for models involving batch (figure 17), group (figure 18) and layer (figure 19) normalization. In the main text we defined the Gini index in terms of the Lorenz curve. It is worth noting that an equivalent definition of the Gini index for n different activities $a_i$, $i = 1, \ldots, n$ in a given layer is given by:

$$G = \frac{\sum_{i=1}^{n} \sum_{j=1}^{n} |a_i - a_j|}{2n \sum_{j=1}^{n} a_j}$$

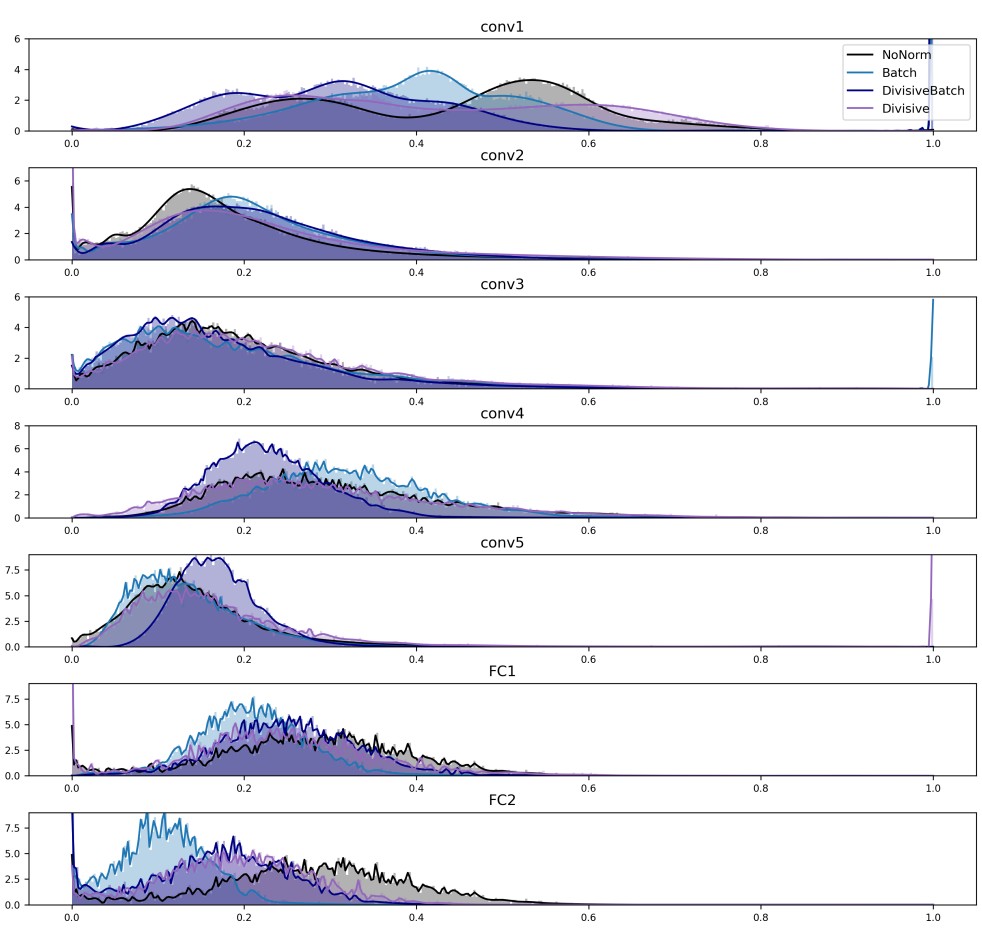

Figure 17: Distribution of activities for the validation set of images at epoch 90, for CIFAR-100 models involving batch normalization, along with the divisive and NoNorm models: Batch (lighter blue), DivisiveBatch (darker blue), Divisive (purple), NoNorm (black).

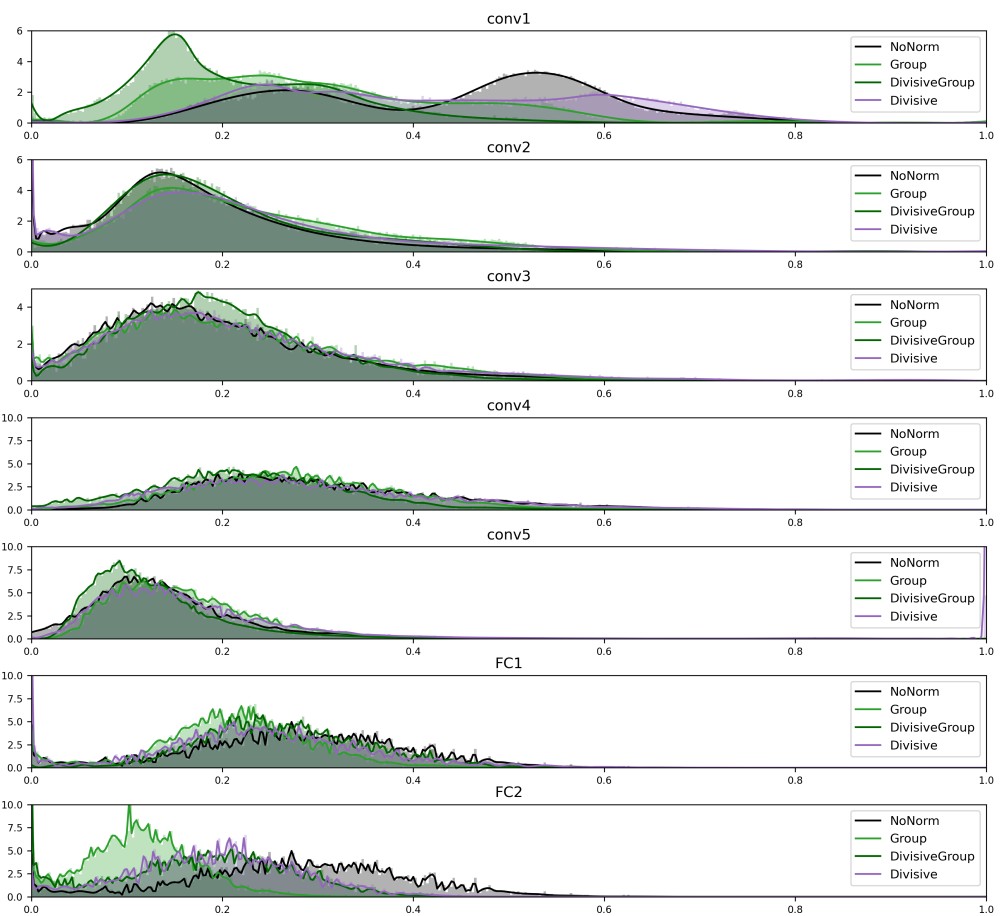

Figure 18: Distribution of activities for the validation set of images at epoch 90, for CIFAR-100 models involving group normalization, along with the divisive and NoNorm models: Group (lighter green), DivisiveGroup (darker green), Divisive (purple), NoNorm (black).

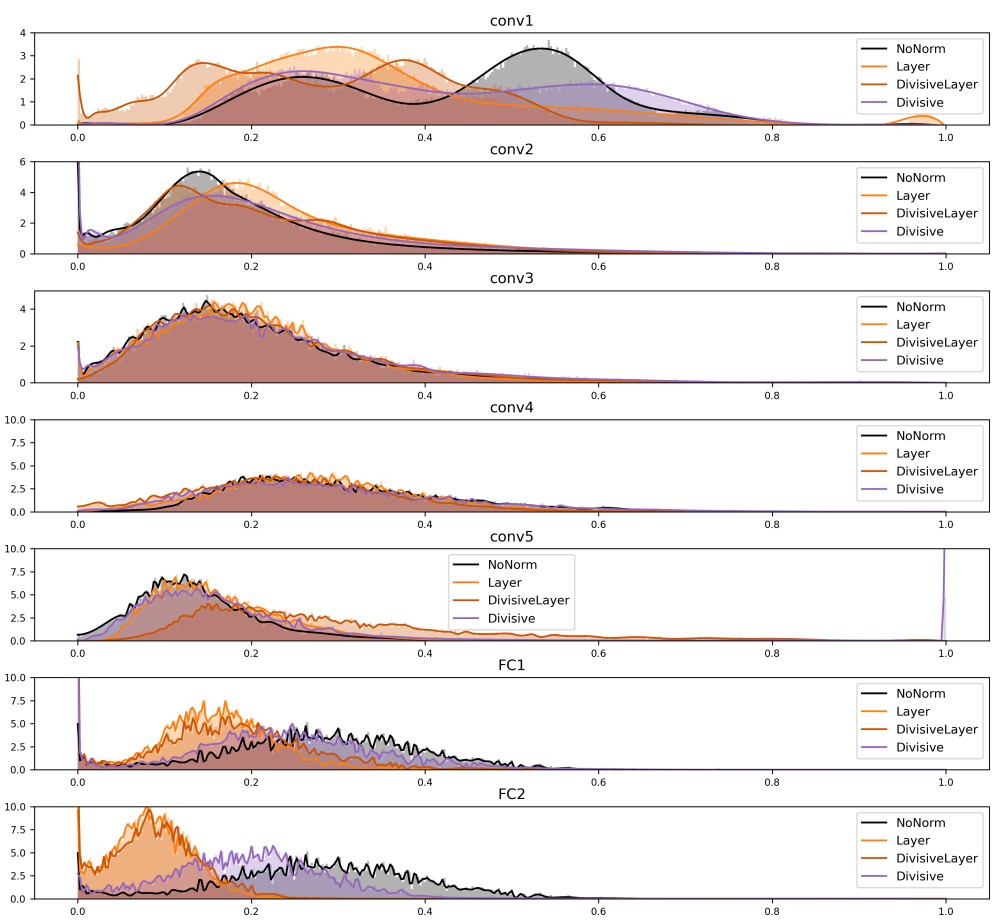

Figure 19: Distribution of activities for the validation set of images at epoch 90, for CIFAR-100 models involving layer normalization, along with the divisive and NoNorm models: Layer (lighter orange), DivisiveLayer (darker orange), Divisive (purple), NoNorm (black).

For each neuron, we calculate the average probability of activation for all images in the validation set for CIFAR-100. We then plot the distribution of these probabilities for each layer. This is plotted for the activations corresponding to the last of: the ReLU; the divisive normalization; and the canonical (batch, group, layer) normalization.

## G  RECEPTIVE FIELDS

We plot the first 25 features and each model to have a closer look at the characteristics of the receptive fields. There is a slightly wider range of intensity for the receptive fields in the Divisive model relative to the NoNorm model. For the form, the values of the receptive fields rang in values from -.96 to 1.18, whereas the latter range from -.79 to +.94. Visually, it is clear the Gabor-like receptive fields are wider field and have lower frequency oscillations than those of the NoNorm model. There are fewer small-scale structures in the Divisive model receptive fields.

Perhaps unsurprisingly from the Fourier analysis, there are still small scale fluctuations and structure from the DivisiveBatch and the Batch models in their receptive fields. Recall the Fourier modes were larger for mid to small scale structure in the combined and canonical models. Interestingly, the Batch model appears to have more amorphous and Gaussian-like receptive fields as opposed to Gabor-like. While there are still some small-scale structures, there are more small-scale structures in the DivisiveBatch mode receptive fields.

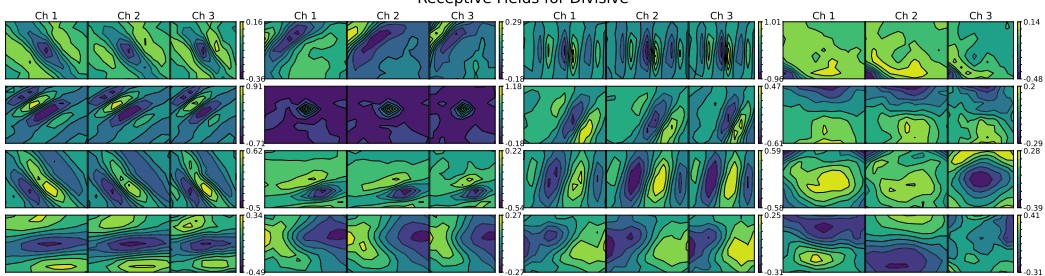

Figure 20: The receptive fields for the first 25 features of the Divisive model. They are clearly wide field Gabor-like receptive fields of various orientations in most channels. Some are more amorphous

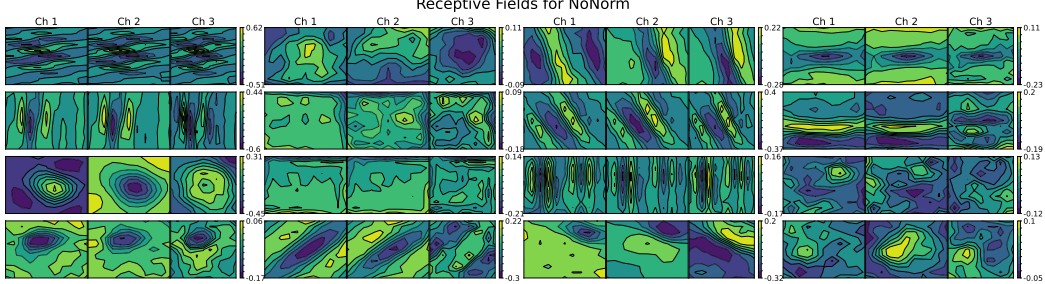

Figure 21: For the first 25 features of the receptive fields in the NoNorm model, there are more small scale fluctuations that are not as prevalent in the receptive fields of the normalized models.

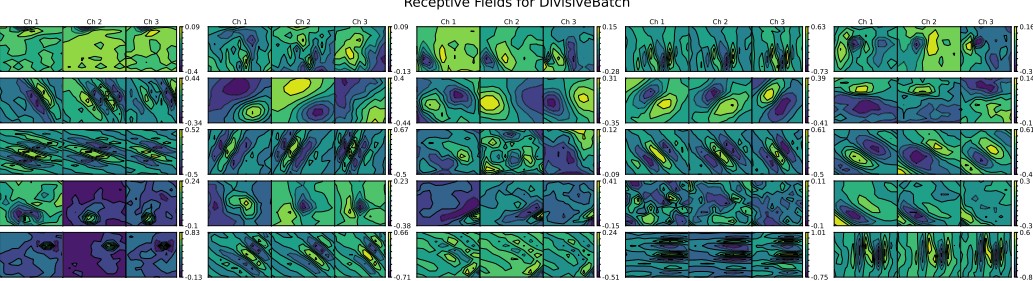

Figure 22: For the first 25 features of the receptive fields in the DivisiveBatch model, there are more small scale fluctuations than those seen in the Divisive model. This could mean that the DivisiveBatch model, while it performs better in accuracy, may not be filtering out small scale fluctuations as well.

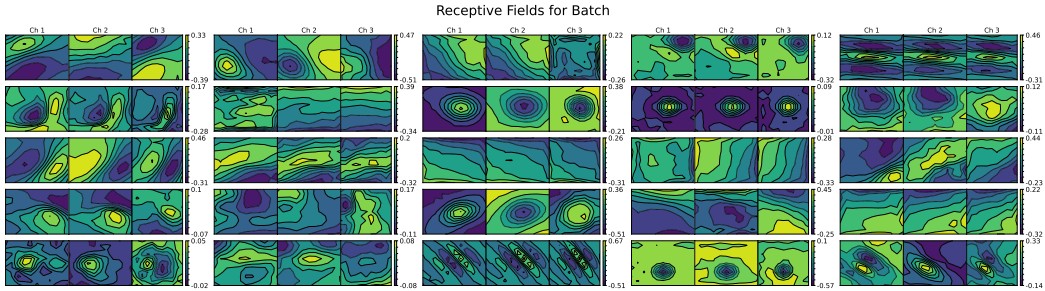

Figure 23: The first 25 features of the receptive fields in the Batch model. There are more rounded receptive fields that appear Gaussian-like. The larger structures could be more large-scale edge detectors.

# H    ADVERSARIAL ATTACKS

Neural network performance can be drastically reduced with very simple modifications to an image known as adversarial examples. Such examples can take two different forms: white box and black box. White box "attacks" are images which have some information (such as the gradients) about the network that is used to create adversarial examples that disrupt the performance. Black box "attacks" have no information about the model. For example, additive Gaussian noise is a black box attack since the noise added to the images is randomly generated regardless of any of the weights or other parameters of the network.

As a beginning exploration of the performance of the various normalizations against adversarial attacks, we tested the models against three different adversarial attacks: two white box attacks, Projective Gradient Descent (PGD) and Fast Gradient Sign Method (FGSM) Goodfellow et al. (2015); and one black box attack, L2 Additive Gaussian Noise (Figs. 24-26). We did this using the FoolBox package developed by Rauber et al. (2020; 2017), The L2 Additive Gaussian noise adds Gaussian noise with a given L2 size (*i.e.*, standard deviation) to the input. FGSM adds noise to the input with a given L2 size that corresponds to the direction of the gradient of the loss function with respect to the data. The PGD method is similar in that it aims to find a perturbation to the pixel that maxmimizes the loss.

While it is hard to generalize from these limited studies, a few results are notable. The Layer model is the most robust of the models in the sense that it is the only model that outperforms the NoNorm model for all three models for (almost) all values of attack strengths (all but the strongest Gaussian noise attack). The models with divisive normalization (except DivisiveGroup) are the most robust against the strongest Gaussian noise. They generally do not perform well against white box models.

In the main text, we discuss performance on Out of Distribution (OOD) images. Because the white box attacks design images targeted to a network's weaknesses based on knowledge of the specific network, we suggest they are not a good test of ability to perform well OOD. A black box attack, being simply a class of OOD images without knowledge of the particular network and its weaknesses, provide a better test, and that is why we discuss the black box attack in the section on OOD performance.

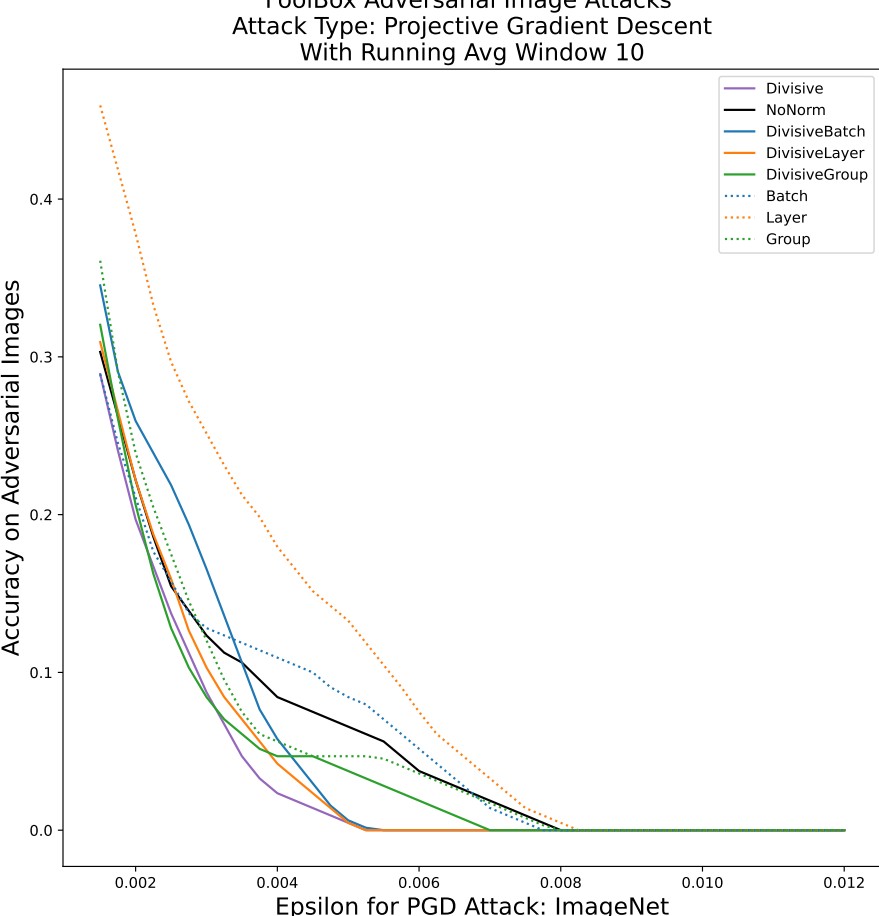

Figure 24: **Projective gradient descent (PGD) attacks**: Using batch sizes of 64, the inputs were given perturbations of L2 norm $\epsilon$ in the direction of the gradient of the loss function with respect to the input. The Layer model is the most robust. Models with divisive normalization tend to be the least robust.

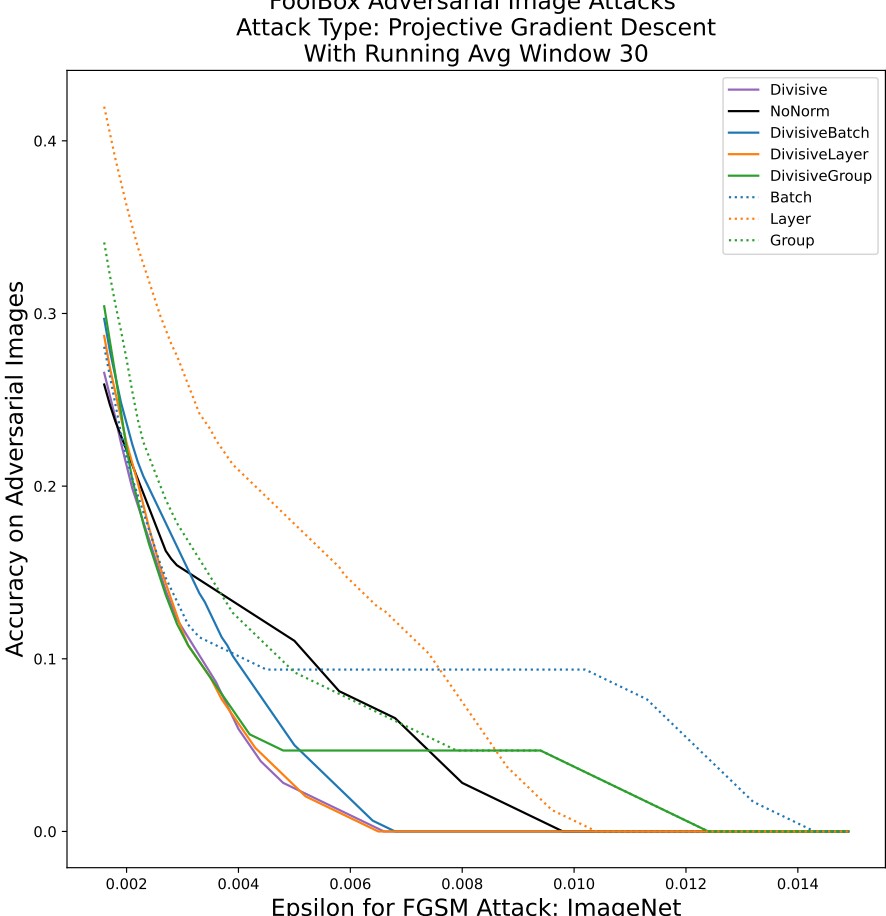

Figure 25: **Fast Gradient Sign Method (FGSM) attacks**: Similarly to the PGD attack, $\epsilon$ measures the strength of the attack. Models with divisive normalization perform poorly relative to NoNorm, except for the DivisiveGroup model for a range of stronger attacks. The most robust models are Layer (weaker attacks) and Batch (stronger attacks).

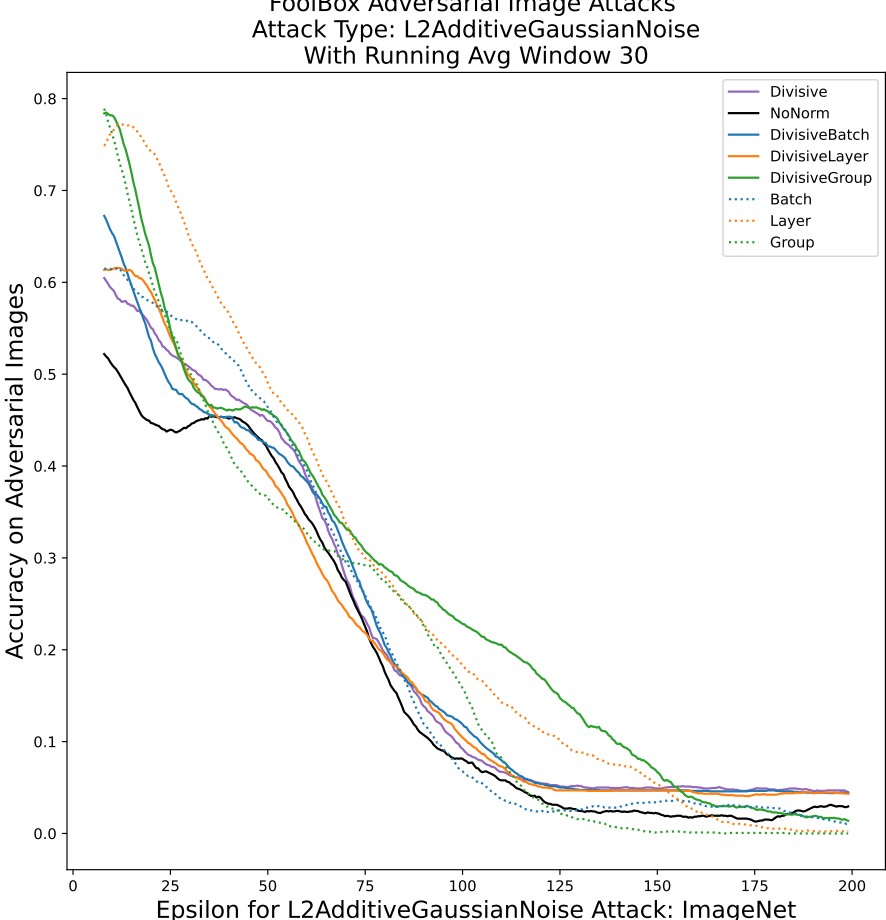

Figure 26: **L2 Additive Gaussian Noise attack**: Using batch sizes of 64, the inputs were perturbed by random Gaussian noise of standard deviation $\epsilon$. For the largest perturbations, most of the models with Divisive normalization (Divisive, DivisiveBatch, DivisiveLayer) are more robust than the other models. For somewhat weaker perturbations, DivisiveGroup is the most robust of the models.

# I SHAPE VS TEXTURE BIAS

We studied shape vs texture bias using the texture-vs-shape package on github developed by (Geirhos et al., 2018). This dataset has 16 shape classes, each with 80 photos. These are a subset of the textures and images used in the stylized ImageNet dataset. They use a style transfer method to impose different textures onto each image. Each shape class has 80 textures for a total dataset of 1280 images. Each image thus has a shape label and a texture label. When a model's inferred category matched the category of either the shape or the texture used in the decision, this was counted as a shape or a texture decision, respectively. The shape bias is the number of shape decisions, divided by the sum of shape and texture decisions. We found that the batch-norm model had a stronger shape bias than no-norm, and, most importantly, that addition of divisive norm to either of these models increased the shape bias (see figure; vertical lines show the mean of model represented with corresponding color).

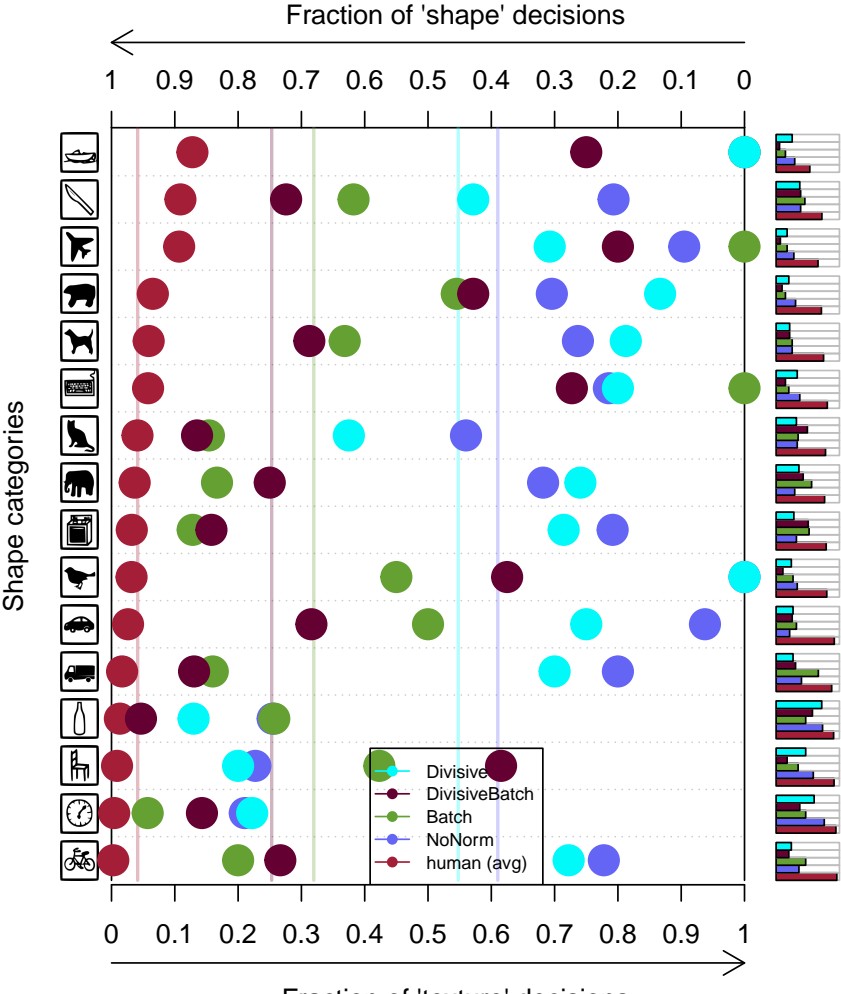

Figure 27:

However, many of the models' decisions were "other" – a category corresponding to neither the shape nor the texture – and the models with divisive normalization had increased "other" decisions (66.8% for Divisive vs. 61.1% for NoNorm; 72.67% for DivisiveBatch vs. 63.75% for Batch). Thus, the models with divisive normalization had an increased shape bias, yet had the same number of or fewer shape decisions out of all categorizations. This gives some hope of divisive normalization and its increased low- or moderate-spatial-frequency power improving shape sensitivity, but does not allow clear conclusions.

## J    CORRELATIONS

For the models trained on ImageNet, we examined the correlations (correlation coefficient; cosine of the angle between the vectors) between features as a function of the distance between them along the feature dimension (Figs. 28-30). This can provide insight into the scale of the impact that Divisive normalization has on developing features, and also reveals some more global effects of the different normalizations. We saw similar behavior for the models trained on CIFAR-100 (not shown).

In any of the models with divisive normalization, there tends to be an anti-correlation induced for nearby neighbors in the first four layers, generally most strongly for the combined divisive/canonical models. That is, features that suppress one another through normalization tend to develop to be anticorrelated. In Layer 5, the DivisiveBatch model shows a weakening, and the DivisiveGroup and DivisiveLayer models a strengthening, of positive correlations between nearby features.

In addition, the combined divisive/canonical models tend to globally decorrelate features, so that correlation is less across all distances than for the other models. This is true in most layers of all the models. For example, for the batch-norm models (figure 28), the DivisiveBatch model has globally weaker correlations than the other models in all layers except layer 1. Furthermore, the models with the canonical normalizations (Batch, Group, Layer) tend to globally have the most positive correlations (for Batch and Group, particularly for layers 2 and 5; for Layer, for layer 5).

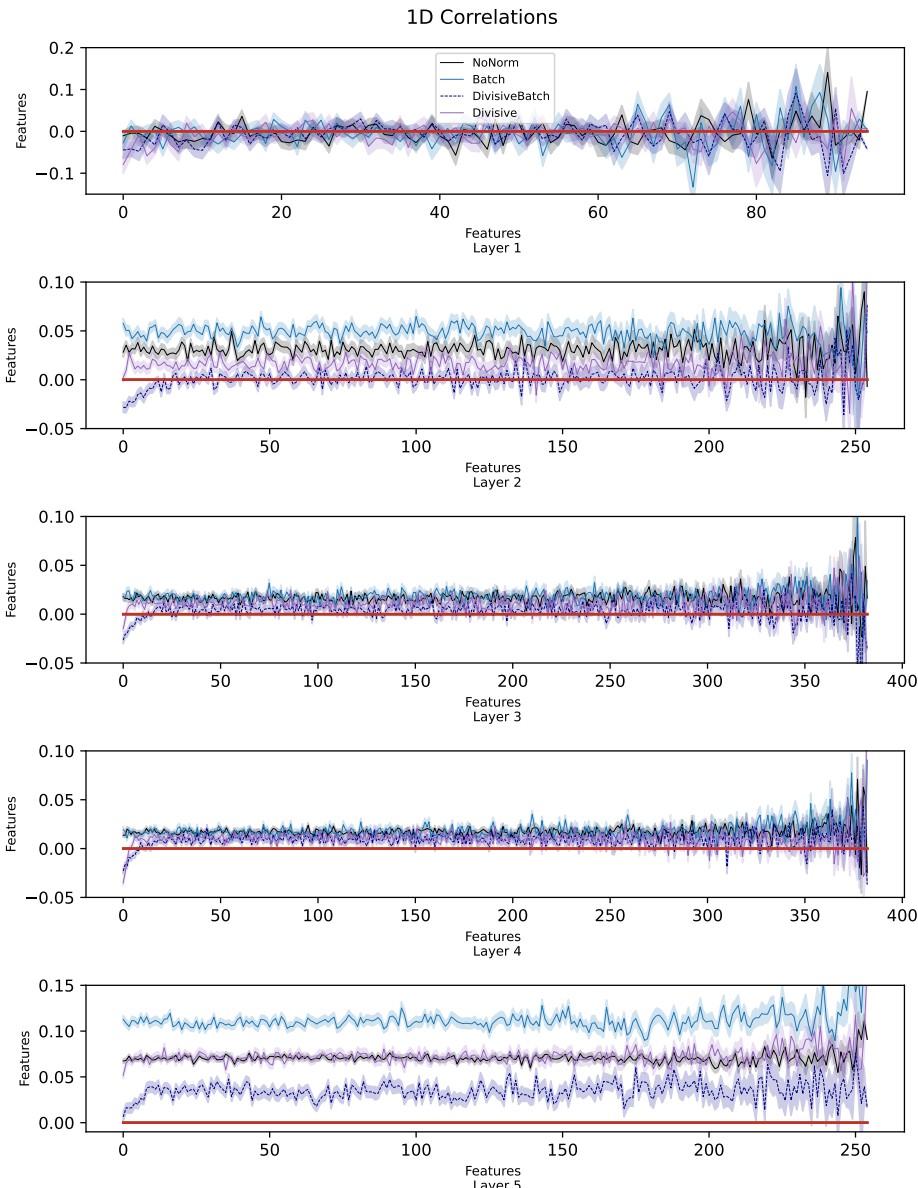

Figure 28: The mean 1D pairwise correlations between features, as a function of the distance between them in the feature dimension, for each convolutional layer, for the DivisiveBatch (Blue,Dashed), Batch (Blue), Divisive (Purple), and NoNorm (Black) models. Shaded regions indicate the standard error of the pairwise correlations. Fluctuations in the correlations grow significantly for further distances as the sample size decreases. This is because many of these layers only have 256 or 384 features, so for correlation distances of 250+ there are very few pairs.

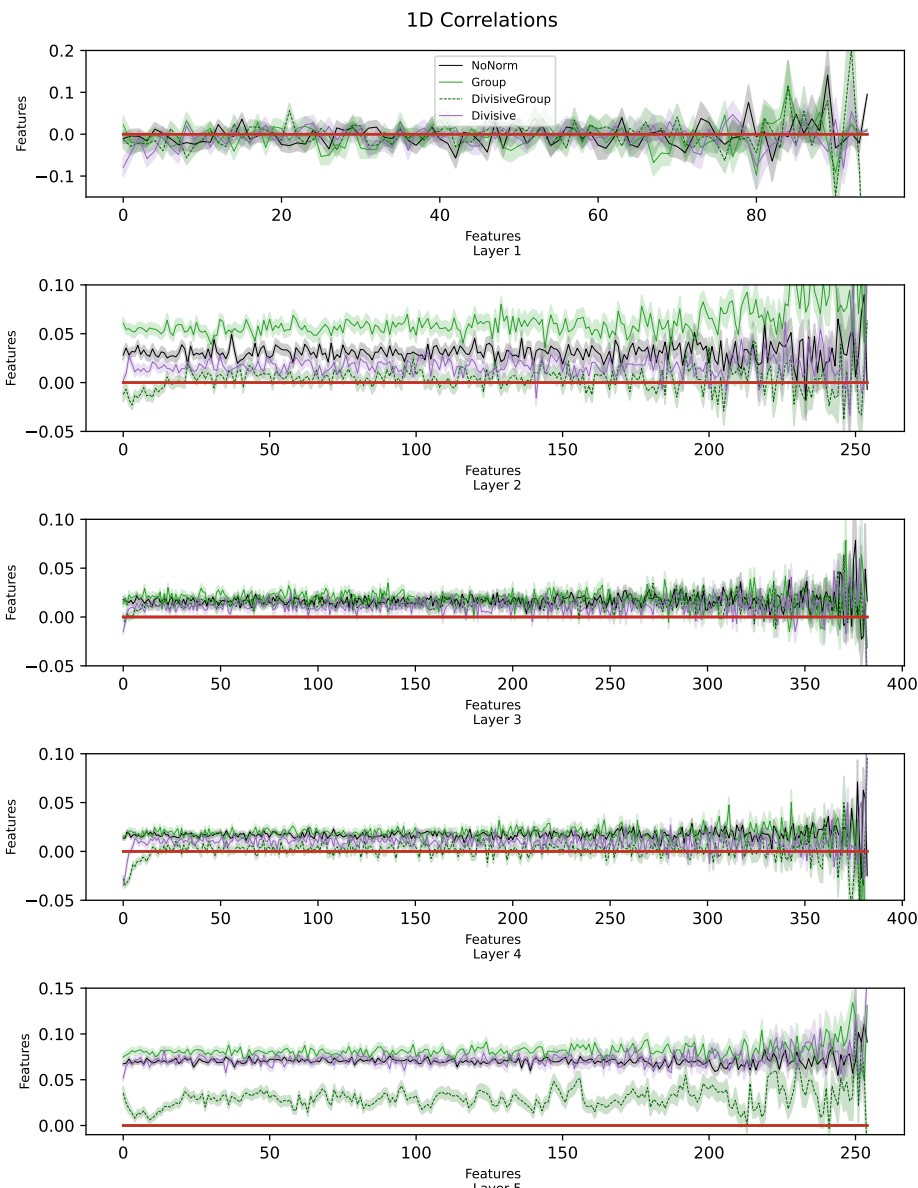

Figure 29: The mean 1D pairwise correlations between features, as a function of the distance between them in the feature dimension, for each convolutional layer, for the DivisiveGroup (Green,Dashed), Group (Green), Divisive (Purple), and NoNorm (Black) models. Otherwise as in figure 28.

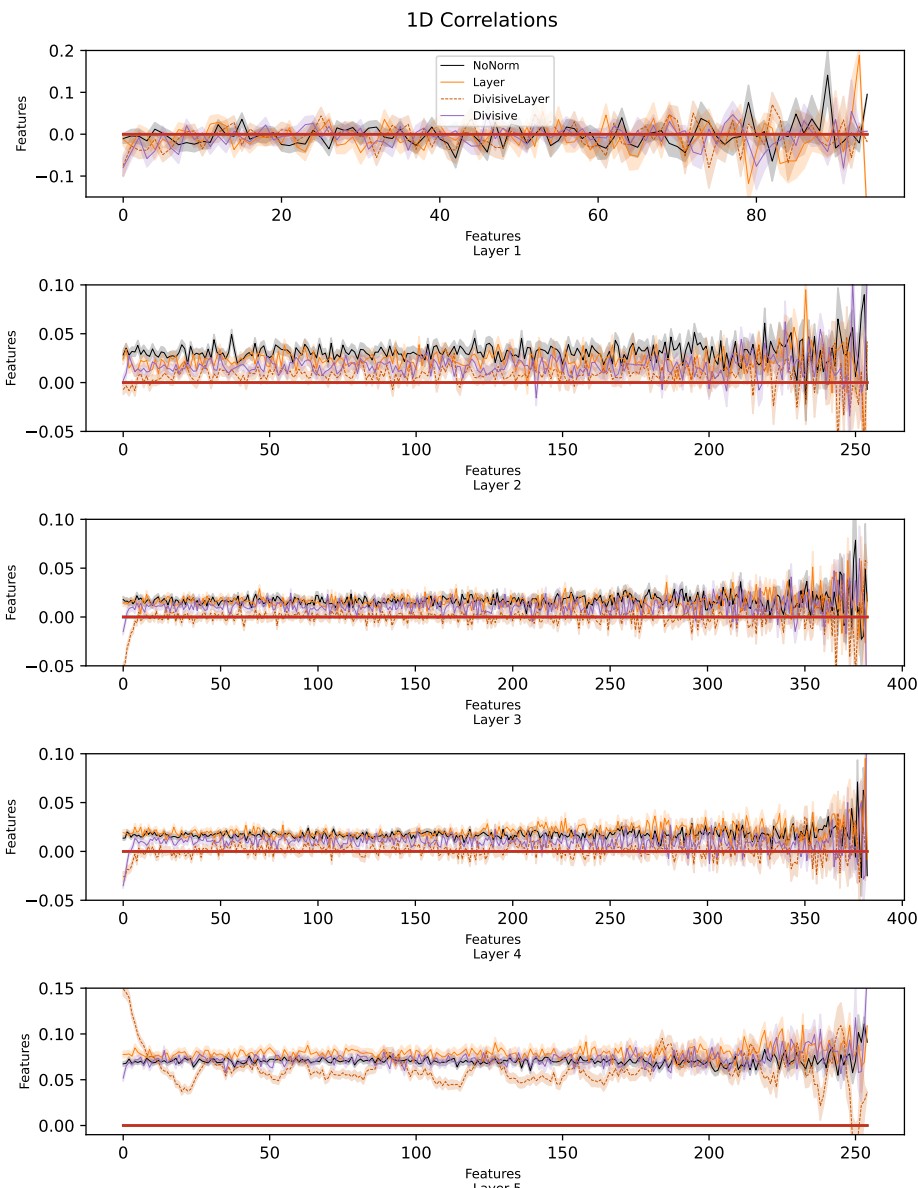

Figure 30: The mean 1D pairwise correlations between features, as a function of the distance between them in the feature dimension, for each convolutional layer, for the DivisiveLayer (Orange,Dashed), Layer (Orange), Divisive (Purple), and NoNorm (Black) models.

## K   ORIENTATION SELECTIVITY

We measured the orientation selectivity of each filter in each color channel in the first layer of the models. We compute an orientation tuning curve by taking the Fourier transform of each 11x11 filter, dividing the 2-D wavenumbers $\mathbf{k}$ into 12 orientation bins (e.g., if the vector $\mathbf{k}$ points between 0 and 15deg or between 180 and 195deg, it goes into the bin of orientation 0 to 15 deg), and calling the response to a given orientation bin the maximum of the amplitude of the Fourier transform for $\mathbf{k}$'s within that bin. This is equivalent to the maximum response to a full-RF sinusoidal grating of any spatial frequency and phase within the given orientation bin. We can then calculate the circular variance ($CV$), a global measure of a tuning curves shape:

$$CV = 1 - \frac{F1}{F0}$$

where F1 and F0 are the amplitudes of the first harmonic and DC of the orientation tuning curve. $CV = 1$ represents no orientation selectivity, while $CV = 0$ is maximum orientation selectivity (nonzero response only to a single orientation).

Figure 31: Violin plots of the NoNorm, Divisive, Batch and DivisiveBatch models indicate a modest drop in the medians and interquartile ranges of the divisively normalized distributions. This difference is significant for DivisiveBatch vs. Batch but not for Divisive vs. NoNorm, see discussion of statistical tests in this Appendix.

To verify whether the CV distributions for the NoNorm vs. Divisive and the Batch vs. DivisiveBatch models are in fact different, we conduct a two sample Kolmogorov-Smirnov (KS) test. The KS test tests whether two different sets of data come from the same distribution. While we do not calculate a statistically significant p-value for NoNorm vs Divisive ($p = .151$), for Batch vs DivisiveBatch, we calculate that the CV distributions are in fact different ($p = .000197$). We also calculate a statistically significant value for Group vs DivisiveGroup ($p = .012$).

Next we use the Mann-Whitney test, which determines whether samples from one distribution are significantly larger than samples from the other distribution. Again, Batch vs. DivisiveBatch and Group vs. Divisive Group are significantly different. Since in both cases the model with

Divisive normalization has the lower median (medians: Batch, .364; DivisiveBatch, .332; Group, .369; DivisiveGroup, .336), we take this to mean that in these two cases the addition of Divisive normalization increases orientation selectivity.

Finally, we calculate the skews of the CV distributions. A negative skew means there is more bulk on the right and a long tail on the left, a positive value implies the opposite (right-skewed). Thus a positive skew suggests a greater bulk of units with low CV (higher orientation selectivity). For Batch and Group, addition of Divisive normalization converted a negative skew to a positive skew, again consistent with Divisive normalization causing a lowering of CV's for these two models. That is the skew for Batch (-.0499) shifts to being positive for DivisiveBatch (.2064) and the skew for Group (-.089) shifts to positive for DivisiveGroup (.304). Thus, there is a shift of the CV values to be smaller with a more positive skew in the divisively normalized models, leading to increased orientation selectivity.

