# OpenReview forum: "Divisive Feature Normalization Improves Image Recognition Performance in AlexNet"
_ICLR.cc/2022/Conference — ICLR 2022 Poster_

### Official Review · Reviewer_nq4A · 2021-10-25

**Correctness:** 3
**Technical Novelty And Significance:** 2
**Empirical Novelty And Significance:** 2
**Recommendation:** 6
**Confidence:** 4

**Main Review:**

Generally, the figure labels are too small and it is difficult to extract the information that they are intended to show.

I think the idea of arranging filters is neat. But this should really be motivated better (e.g. discuss topographic ICA or recent work on topographic CNNs). Also, it would be interesting to see the learned topography.

The brief conclusion section about robustness is interesting but feels ad-hoc.

The appendix looks like a giant dump of plots and additional information. While there is no constrant on appendix length, it would still be nice to arrange and filters this better, so that a reader looking for specific details could find them more easily. It is also unreasonable to expect that reviewers go through >20 pages of appendix to verify its correctness.

Comments:
- Why study only AlexNet and no newer CNN architecture?
- The contributions list is rather a list of findings
- I do not understand the first sentence of section 2
- I think Balle & Simoncelli's GDN (ICLR, 2017) model might fit well into the related work
- What performance do you see with the original AlexNet, i.e. keeping LRN?
- Other works (e.g. Burg et al.) have shown that the power in eq. should also be learned. Why did you fix it to 2?
- Section 4: by 'trial', do you mean 'seed' on Cifar100?
- 'Receptive fields and Fourier power': The second sentence starts with 'therefore', this seems like a non sequitur.
- Figure 2 is illegible, there is too much information and it is not at all clear what to look at or what this is supposed to show.
- You write that a model with higher performance also has higher capacity, this seems wrong in the general case.
- In the sparsity section you write 'respond better', what does that mean? Higher response value given fixed contrast or something like that?
- The Gini index is a little confusing, why don't you just measure entropy? This would seem more easy to follow for an ML audience.
- Fig5 How would these curves look like for Cifar100 across seeds? I.e. is this a significant observation. Why did you use only 100 images and what do you mean by least correlated classes? Also, more generally, what have we learned from looking at these metrics?
- Error bars for Fig6?

Small suggestions:
NN's -> NNs

**Summary Of The Paper:**

The authors study the effect of divisive normalization on AlexNet. They show that, when combined with standard normalization schemes, it increases performance. They also investigate the filter shapes, manifold capacity and (adversarial) robustness of the learned representations.

Following the authors' response I have increased my score form 5 to 6.

**Summary Of The Review:**

The paper explores an interesting but not novel idea. The analysis is extensive, but one is left wondering what we have learned from the study. More generally, there is a mixture of ideas and analyses which are interesting but somehow not well connected in this work.

---

> ### Author Response · Authors · 2021-11-23
> **Part 1 of Response:**
>
> We thank the reviewer for their time spent reviewing our paper. We are happy the reviewer finds the ideas we are exploring interesting. We incorporated their suggestions to our updated manuscript. Below we provide responses to the comments, in particular:
>
> $\textbf{Main Review:}$
> $\\textbf{I think the idea of arranging filters is neat. But this should really be motivated}$
> $\\textbf{ better (e.g. discuss topographic ICA or recent work on topographic CNNs).}$
> $\\textbf{ Also, it would be interesting to see the learned topography.}$
>
> --The learned topography is illustrated in the sets of 16 adjacent RFs shown in Fig 4 in the main paper and in Appendix G in the Supplement, and also in the plots of average correlation vs distance shown in Appendix J. But we don’t think the motivation in terms of topographic ICA or topographic CNNs exactly fits, as we are not explicitly trying to topographically group together, for example, RFs that share particular features. Furthermore, the plots of correlation vs. distance show that, if anything, the normalization leads to an “anti-topography” -- nearby filter pairs become less similar than farther apart, non-interacting filter pairs. $\\newline$
>
> --Our motivation is not to study a form of topographic CNN, but to study the divisive normalization seen biologically. Because the RFs (filters) of the neurons that are normalizing each other develop along with the normalization in our models, and similarly biologically the two must co-evolve or co-develop biologically, there is a need for a topological arrangement that constrains which developing RFs can potentially interact, but this does not imply a topographic organization of features. Accordingly, with respect, given the severe space limits of the paper and the large number of topics to be addressed, we have not chosen to add a discussion of topographic CNNs or topographic ICA. $\\newline$
>
> $\\textbf{The brief conclusion section about robustness is interesting but feels ad-hoc.}$
>
> We have rewritten this section of the discussion.
>
> $\textbf{Comments}$:
>
> $\\textbf{Why study only AlexNet and no newer CNN architecture?}$
>
> Primarily for simplicity. We are comparing many different models (i.e., choice of canonical normalization and presence or absence of DN) and in some cases different tasks (CIFAR vs ImageNET). These must all be compared on a common architecture for meaningful comparisons, and for simplicity (and given computational resources) we restricted to doing these comparisons only for one architecture. Another motivation is that the layers of AlexNet show a reasonable correspondence to the hierarchy of areas in the primate visual stream, whereas the large numbers of layers in some other architectures lack such a simple correspondence; it made sense to first study normalization in layers that seem to have a biological analog. A third consideration is that it took some work to find proper conditions for stable learning with DN in all five layers, and would have taken still more work to find such conditions for models with many more layers, although we did some initial exploration of other architectures.  $\\newline$
>
> $\\textbf{I think Balle and Simoncelli's GDN (ICLR, 2017) model might fit well in related work}$
>
> We have added a reference to this in section 2.  $\\newline$
>
> $\\textbf{What performance do you see with the original AlexNet, i.e. keeping LRN?}$
>
> On the ImageNet dataset, there is roughly a 1. point improvement in validation accuracy and about 1.7 point improvement on training accuracy when looking at the NoNorm model with the Pytorch LRN vs just NoNorm. DN performs better than the Pytorch LRN. We have updated the table to include these numbers, which are: for validation accuracy at epoch 90 for imagenet models, NoNorm 56.49; NoNorm + PytorchLRN 57.55; Divisive 59.39.  $\\newline$
>
> $\\textbf{Other works (e.g. Burg et al.) have shown that the power in eq. should}$
> $\\textbf{ also be learned. Why did you fix it to 2?}$
>
> In the original paper on divisive normalization (Heeger, 1992) the power 2 was also used. In other studies of power law input output functions (e.g. Rubin et al 2015), it appeared that the main determinant of behavior was that the power be greater than one (i.e. the power be expansive). We cannot rule out that it would have helped to learn this, but it is a parameter that we kept fixed for simplicity to focus on the other parameters.  $\\newline$
>
> We note that, while Burg et al. did learn the power, and showed the learned distribution of powers, almost all >1, they did not address whether this learning helped their model to better optimize its cost function, i.e. the ability to account for responses of V1 neurons. $\\newline$

---

> > ### Author Response · Authors · 2021-11-23
> > **Part 2 of Response:**
> >
> > $\\textbf{Section 4: by 'trial', do you mean 'seed' on Cifar100?}$
> >
> > Yes  $\\newline$
> >
> > $\\textbf{The contributions list is rather a list of findings}$
> >
> > We have rewritten that section.  $\\newline$
> >
> > $\\textbf{I do not understand the first sentence of section 2}$
> > $\\newline$
> >
> > As we had noted in section 1, divisive normalization is a phenomenological description of neural responses, not a circuit model. The first sentence of section 2 describes a circuit model that produces the responses that are described by divisive normalization. We have rewritten the sentence to try to make this more clear.  $\\newline$
> >
> > $\\textbf{Figure 2 is illegible, there is too much information and it is not at all clear what to }$
> > $\\textbf{look at or what this is supposed to show.}$
> > We have replaced Fig. 2 with the figure below, in which we averaged the power across the three color channels. We hope this simplifies the plot enough to make it more readable.  $\\newline$
> >
> > $\\textbf{Link for plots!}$
> > LINK : https://imgur.com/a/plmTE5i
> >
> > $\\newline$
> > $\\textbf{You write that a model with higher performance also has higher }$
> > $\\textbf{capacity, this seems wrong in the general case.}$
> > $\\newline$
> >
> > We agree that the manifold capacity and the accuracy have technically different definitions. However, they are both standard measures of linear separability. Furthermore, the manifold capacity metric has been shown to have a strong correlation with accuracy in recent studies (Dapello et al, Stephenson et al). Developing a theory that establishes a formal connection between accuracy and representational geometric properties such as each category manifold’s dimension & size would be an interesting direction for future study, but outside the scope of this paper. We have rewritten this statement to make it more accurate. $\\newline$
> >
> > $\\textbf{In the sparsity section you write 'respond better', what does that mean?}$
> > $\\textbf{ Higher response value given fixed contrast or something like that?}$
> > $\\newline$
> >
> > We were referring to the Zhuang et al. 2017 paper, which compared the responses to paired naturalistic/non-naturalistic textures derived from the same underlying natural texture image. Both have the same first- and 2nd-order statistics (e.g., contrasts) as the underlying image, but the non-naturalistic textures have Gaussian statistics whereas the naturalistic stimuli also share higher-order statistics with the underlying image. They found that units in higher layers of their network strongly tended to respond more strongly to the naturalistic texture than to its paired non-naturalistic texture. They further found, by using various experimental manipulations to change the sparseness of activations that developed in given layers, that this preference for the naturalistic texture was strongly correlated with the sparseness of activation. This is what we are referring to when we say “Both higher areas of visual cortex and higher layers of AlexNet and other deep nets trained for visual object recognition respond better to stimuli with naturalistic statistics than those without them (Zhuang et al 2017), and in the deep net higher layers this property is strongly correlated with sparsity of firing and not with any other tested property (Zhuang et al 2017).”
> >
> > $\\textbf{“The Gini index is a little confusing, why don't you just measure entropy?”}$
> > $\\newline$
> >
> > --The entropy would not immediately tell us anything about the sparsity of the network. If one bins the data, the entropy is unchanged by permuting which bins correspond to which activity level, whereas the GINI index specifically examines the portion of the distribution that is in higher vs. lower activity levels (analogous to the portion of the total wealth that is in those with the highest wealth). Thus the GINI index, but not the entropy, speaks to the sparsity of the activity -- i.e., examining whether only a small number of units have high activities. $\\newline$

---

> > > ### Author Response · Authors · 2021-11-23
> > > **Part 3 of Response**
> > >
> > > $\\textbf{“Fig5 How would these curves look like for Cifar100 across seeds?}$
> > > $\\textbf{ I.e. is this a significant observation. Why did you use only 100 images }$
> > > $\\textbf{and what do you mean by least correlated classes? Also, more generally,}$
> > > $\\textbf{ what have we learned from looking at these metrics?”}$
> > > $\\newline$
> > >
> > > --Figure 5 selects 100 images from the least correlated classes 5 different times, using different seeds for each selection. Error bars are shown, though small, corresponding to the standard error over the 5 selections.
> > > Least correlated classes refer to classes with lowest correlation between their centroids, so that the selection of the data is consistent with the assumptions of the theory for the capacity metric (Cite: Chung et al, 2018). The choice of the least correlated classes here follows the list of classes provided in the prior work (Cite: Cohen and Chung et al, 2020). $\\newline$
> > >
> > > -- Again, as in our response to reviewer 1, we are learning about how the geometry of the object manifolds, a key aspect of the learned representations, is shaped by learning. Furthermore we are learning the surprising fact that DN reduces manifold capacity in early layers and that this leads to increased capacity and performance in the final layer, a phenomenon that deserves future examination.$\\newline$
> > >
> > > $\\textbf{“Error bars for Fig6?”}$
> > > $\\newline$
> > >
> > > --Here, we formed the distribution of activations of all units in a given layer across all of the images in the validation set, and computed the Gini index. Since there is only one distribution, there is only one Gini index, with no error bars. Given the huge number of sampled points (~50,000-400,000 units per layer times 20,000 images in the validation set), the Gini index should be quite precise. Alternatively, we could have formed many smaller distributions, for example, one for each image, obtained a Gini index for each smaller distribution, and then found mean and error bars over this distribution of Gini indices. However, this is not the approach we took.
> > >
> > > $\\textbf{The reviewers requests that we make the appendix more readable.}$ $\\newline$
> > >
> > > --We have now included a table of contents for the Appendix. We have also endeavored to ensure that every section of the Appendix is specifically referred to at a particular point in the main paper, so that the reader will be pointed to the relevant parts of the Appendix as they read the paper.
> > >
> > > $\\textbf{Citations for above discussion:}$
> > >
> > > SueYeon Chung, Daniel D. Lee, and Haim Sompolinsky. Classification and geometry of general perceptual manifolds. Phys. Rev. X, 8:031003, Jul 2018. doi: 10.1103/PhysRevX.8.031003. URL https://link.aps.org/doi/10.1103/PhysRevX.8.031003.
> > >
> > > Cohen U DiCarlo JJ Sompolinksy H Chung S, Dapello J. Separable manifold geometry in macaque ventral stream and dcnns. In Computational and Systems Neuroscience, 2020.
> > >
> > > Cory Stephenson, Jenelle Feather, Suchismita Padhy, Oguz Elibol, Hanlin Tang, Josh McDermott, and SueYeon Chung. Untangling in invariant speech recognition. In NeurIPS, 2019.
> > >
> > > C. Zhuang, Y. Wang, D. Yamins, and X. Hu. Deep Learning Predicts Correlation between a Functional Signature of Higher Visual Areas and Sparse Firing of Neurons. Front Comput Neurosci, 11:100, 2017.
> > >
> > > D. J. Heeger. Normalization of cell responses in cat striate cortex. Vis. Neurosci., 9:181–198, 1992.

---

> > > > ### Comment · Reviewer_nq4A · 2021-11-24
> > > > **Response to response**
> > > >
> > > > I thank the authors for their thorough response. I understand now the motivation for using topographic filters as a way of reducing the number of learnable parameters in the normalization pool. It would be great if that explanation could be included in the paper right were the topography idea is introduced to make it clear why you are doing this. Otherwise, I am happy about the revisions and will increase my score accordingly.

---

> > > > > ### Author Response · Authors · 2021-11-24
> > > > > **Thanks**
> > > > >
> > > > > Thank you. We can't currently edit the paper but, if it is accepted, we can edit to clarify the motivation for topography in the final paper.

---

### Official Review · Reviewer_h6d9 · 2021-10-29

**Correctness:** 3
**Technical Novelty And Significance:** 3
**Empirical Novelty And Significance:** 3
**Recommendation:** 8
**Confidence:** 4

**Main Review:**


The results show on the one hand a slight increase in performance. Interestingly and surely more significant in terms of the scope of this paper are the structural changes that are made in the network and in particular the increase in the level of sparsity of the activity in the different layers as well as on the shape of the receptive fields.

In general the paper is very well presented and the results are very convincing. I have some minor points that I detail below

- The abstract in particular is too long and should be cut in half.

- In each layer, the channels are arranged topologically along a line. Some computational neuroscience models especially those representing orientations are arranged according to a circular typology. Have you tested this option?

- In figure 1, what happens during learning around epoch 30?

- In figures 2, 5, 6 the characters describing the X and Y axes are too small to be readable.

- In figure 4 and four the receptive fields could be shown in color in order to show all the receptive fields higher.

- Is the choice of the AlexNet model important for the generality of the results or does this also apply to other networks such as VGG?

- Finally, the weights of the normalization model are learned during the training. On the other hand, you do not analyze the weights that are obtained at convergence and whether they can be interpreted in terms of the relations between the receptive fields, as can be done in neuroscientific models like those of Schwartz and Simoncelli.



**Summary Of The Paper:**

In this paper, the authors propose to study the influence of a feature of biological neural networks on a classical deep learning network. In particular, the authors are interested in including a divisive normalization like the one characterized in the primary visual cortical era and to see the influence of this mechanism on the performance of the AlexNet network.


**Summary Of The Review:**

These findings could help explain a largely unanswered question in neuroscience, which is to understand why divisible normalization, which is first and foremost a phenomenological heuristic, exists in biological networks.

---

> ### Comment · Reviewer_h6d9 · 2021-11-22
> **no more comment**
>
> As there was no answer to my comments, I have updated my evaluation accordingly.

---

> > ### Author Response · Authors · 2021-11-22
> > **Responses are coming**
> >
> > We thank reviewer h6d9 for their comments and ask for the reviewer's patience. We will be uploading our responses and the revised paper by the deadline tonight. It is possible that we misunderstood the customary ICLR procedures, but we have been waiting to finish our revisions before uploading our responses, since our responses refer to revisions we are making. Thank you.

---

> > > ### Comment · Reviewer_h6d9 · 2021-11-22
> > > **got it**
> > >
> > > ok - there was no intention to pressure, sorry for that. but my understanding is that the discussion period is over tonight:
> > >
> > > Author / Reviewer / AC Discussion Period Ends 	Nov 22 '21(Anywhere on Earth)
> > >
> > > (from the iclr.cc site)
> > >
> > > and that the window of opportunity in which I will be able evaluate your response and potentially to change my score will be short. please tell me if I am wrong.

---

> > > > ### Author Response · Authors · 2021-11-22
> > > > **Deadlines**
> > > >
> > > > This is confusing. We have been going by the recent email of Nov 19 with subject "[ICLR 2022] Reminder of Discussion Phase Transition" which stated "The deadline to update your ICLR draft is 11:59pm anywhere on earth on Nov 22. After that, we will enter the final stage of the discussion, when you can continue to communicate with AC/reviewers on OpenReview, but won’t be able to update the draft." But as you point out, that is not what the web site says. I hope the email is correct.

---

> > > > > ### Author Response · Authors · 2021-11-22
> > > > > **Discussion period will continue 1 more week**
> > > > >
> > > > > I emailed ICLR to ask about the discrepancy between what the email said and what it said on the website. I got a reply from Yan Liu, yanliucs@usc.edu, saying "Both are correct. The first two stages of discussions (which ends on Nov 22nd) are focused on authors' feedback (including updating the draft) and public comments. The last stage of the discussion (Nov 22nd-Nov 29th) is focused on reviewer/AC discussions and they might ask additional questions to the authors, so authors are expected to answer questions if any."

---

> > > > > > ### Comment · Reviewer_h6d9 · 2021-11-23
> > > > > > **end of discussion phase**
> > > > > >
> > > > > > yes, there is one more week but "After (nov 22), the authors will not be able to update the submission draft."
> > > > > >
> > > > > > It is usual that authors and reviewers exchange in the first phase, for instance in case there are misunderstandings. This allows authors to adapt their paper accordingly. But this is not a necessity.
> > > > > >
> > > > > > Be assured that your corrected paper will be reviewed as any other paper (just that you won't be able to amend it if reviewers have further constraints) and in particular that the score will be updated accordingly.

---

> > > ### Comment · Reviewer_h6d9 · 2021-11-28
> > > **thanks for your answers**
> > >
> > > responses were clear and concise, and while they came late for a proper discussion,  they allowed to alleviate some doubts I initially had. the paper is of broad interest for the community and I have raised my evaluation accordingly to 8.

---

> ### Author Response · Authors · 2021-11-23
> **Response:**
>
> We thank the reviewer for their time spent reviewing our paper. We appreciated their finding that our work could help explain a largely unanswered question in neuroscience. We incorporated their suggestions to our updated manuscript. Below we provide responses to the comments, in particular:
>
> $\\textbf{“The abstract in particular is too long and should be cut in half.”}$
>  $\\newline$
>
> --We shortened the abstract. p.s. The abstract does not appear to be changed on the OpenReview page, and we do not see how to change it. But it is changed in the paper. $\\newline$
>
> $\\textbf{“Some computational neuroscience models especially those representing orientations }$
> $\\textbf{ are arranged according to a circular typology. Have you tested this option?”}$
>
> --The number of channels in a layer varies from 96 to 384 across channels, and the learned lambdas (space constant of the exponential normalization filters) are in the range 2 - 10. Thus the boundary effects of having edges (line representation) rather than no edges (circular representation) will affect only a tiny portion of the channels. $\\newline$
>
> --We did try circular topologies at some points in time. Largely for historical reasons as our methods converged we ended up using a line topology, but in cases when we could compare the two we did not observe obvious differences. We can examine circular topologies in a final paper. $\\newline$
>
> $\\textbf{“In figure 1, what happens during learning around epoch 30?”}$
>
> --We have a learning rate scheduler that reduces the learning rate by a factor of 10 every 30 epochs. We now note this in the Figure 1 caption. $\\newline$
>
> $\\textbf{“Is the choice of the AlexNet model important for the generality of the results }$
> $\\textbf{ or does this also apply to other networks such as VGG?”}$
>
> --We initially did some limited exploration of Resnet 34 and VGG, when we were initially exploring conditions to get stable learning. We first found conditions for reliable learning in AlexNet. Because the limited number of layers of Alexnet seems to better correspond to the limited number of areas in the primate ventral stream, whereas the many layers of the other models are harder to interpret biologically, we then confined our attention to AlexNet. We have no reason to think DN would not generalize to the other architectures. $\\newline$
>
> $\\textbf{“You do not analyze the weights that are obtained at convergence and whether}$
> $\\textbf{ they can be interpreted in terms of the relations between the receptive fields,}$
> $\\textbf{ as can be done in neuroscientific models like those of Schwartz and Simoncelli.” }$
>
> -- If one looks at the correlations for the weights of nearby neighbors (see new section "Feature Correlations", and Appendix J), there is a pronounced decorrelation or anti-correlation induced by divisive normalization for nearby neighbors along the feature dimension. This decorrelation is generally strongest for the combined divisive/canonical models, which in some layers decorrelate the weights for all distances, although most strongly for nearby neighbors. This is shown and discussed in Appendix I. $\\newline$
>
> --The models of Schwartz and Simoncelli look at normalization between idealized RFs, eg Gabor filters or similar. They look at which RFs should normalize which for normative reasons, but no one has been able to directly measure this in actual animals -- the best one can do is fit models that try to account for neural responses. Accordingly a comparison to “who normalizes whom” in neurobiology cannot be done, because we do not know the neurobiological answer. In addition, while Gabor filters or similar reasonably approximate V1 receptive fields in the monkey, the mouse, for example, has RFs that are much messier (Walker et al., 2020 -- “inception loops” paper), as do the RFs in our networks, which simultaneously learn both the filters and the normalization parameters. Even if one adopts the normative principles of S&S, it is not necessarily clear how they should apply to these messy RFs. Because these RFs are not simply characterized, we found the correlation analysis described in the previous paragraph to be the best approach we could find to answer “who normalizes whom” in our networks. $\\newline$
>
> $\\textbf{Citations for above discussion:}$
>
>  O. Schwartz and E. P. Simoncelli. Natural signal statistics and sensory gain control. Nat Neurosci, 4:819–825, 2001.
>
>  Walker, E.Y., Sinz, F.H., Cobos, E. et al. Inception loops discover what excites neurons most using deep predictive models. Nat Neurosci 22, 2060–2065 (2019). https://doi.org/10.1038/s41593-019-0517-x

---

### Official Review · Reviewer_Qotw · 2021-11-02

**Correctness:** 3
**Technical Novelty And Significance:** 4
**Empirical Novelty And Significance:** 3
**Recommendation:** 8
**Confidence:** 5

**Main Review:**

The incorporation of biological nonlinearities in artificial networks is interesting both to enrich architectures in machine learning and to better study computational principles in neuroscience. Therefore, studies like this are (and will be) of interest for the ICLR audience.

I advocate for the publication of this work in the conference when the authors address/comment on the following concerns:

A. LIMITATIONS OF THE CONSIDERED NORMALIZATION AND TASK OBSCURE THE MESSAGE: clarify discussion!

While I trust the reported results (given the use of current-and-standard analysis tools, e.g. Chung18), I feel the 2%-improvement message (as it is written now) can be misleading given the restricted nature of the normalization and the specific training task. The same is true about the evolution of sparsity or the decorrelation between the responses: all depends on the details of the selected interactions in the normalization. The modest results may give the impression that biological normalization is not that different from the conventional normalizations already in use, and hence it may not be worth pursuing that path.

I am persuaded that (1) using more general (and "explainable") divisive normalization schemes and (2) considering visual tasks that require the specific adaptation that can be obtained from "explainable" relations between "explainable" features, would certainly make a big quantitative difference. And the results would be qualitatively "explainable".

Regarding (1), including interaction between neurons tuned to different spatial locations (which is not considered in this work) is crucial to compute local illumination and local contrast, and these are fundamental to make proper classification decisions in scenarios with uncontrolled illumination, shadows or fog. On the other hand, interaction between neurons tuned to similar textures (similar in spatial frequency and orientation) is crucial to increase the statistical independence among responses. This will lead to better information transmission and eventually better classifications.
Normalization between sensors tuned to similar features should also happen at higher abstraction levels(e.g. normalization of similar mid-level primitives or similar objects). This certainly happens in the human visual system (see Webster JoV 2011 on adaptation and masking). In this regard, a major problem of the considered divisive normalization is that it takes a limited line topography in the features. By doing so, in order to get an effective normalization, the evolving features should organize themselves along that line in certain order to let the Gaussian neighborhoods of evolving width do their job. If this order is not enforced in anyway, it is unlikely that it happens leading to a suboptimal normalization.
A possible way to solve this could use an analysis of the emerging receptive fields and a reconfiguration of the Gaussian neighborhoods in the normalization depending on the observed similarities of the receptive fields.

Regarding (2), instead of using generic databases, one could control the training conditions to enforce controlled invariances (color constancy, contrast invariance, invariances to simple geometrical transforms, etc.). These tasks would favour specific masking interactions between specific sensors whenever these sensors are identified as stated above. As a result, (i) the appropriate neighborhoods could be easier to learn, and (ii) the function of the final interaction kernel would be easier to understand.

The limitations of the considered interaction kernel in the Div.Norm. and the limitations of the training sets (or tasks) should be more explicitly stated in the discussion. It has to be acknowledged that results could be more compelling and explainable if these limitations are solved. I am not suggesting to solve them now, since there is no time in the reviewing process, but this should be included in the discussion since it may inspire further work.

B. IS THE FINAL DIVISIVE NORMALIZATION BIOLOGICALLY SENSIBLE?

Biological plausibility is central for cross fertilization from machine learning to neuroscience. The considered normalization is certainly inspired by the biological models, but, in the end, do the final results make biological sense?. This is not checked in the current work in any way.

This question is not independent from previous comment A: for instance, Martinez et al. Front. Neurosci. 2019 showed that normalization models with limited architecture (i.e. simplistic neighborhoods missing relevant interactions) may lead to good performance in perceptually sensible tasks, but they may fail to reproduce classical psychophysics. In this mutidisciplinary (machine learning-and-neuroscience) context, it is relevant to check the biological plausibility of the models that emerge after training.

A possibility would be reproducing human image quality opinion from the final stage of the network before the dense classification layers (following Zhang IEEE CVPR 2018 or Hepburn IEEE ICIP 2020). Do the models with Divisive Normalization provide a more accurate reproduction of human opinion?.
Another option (following Martinez Front. Neurosci. 19) is checking classical psychophysics such as the Contrast Sensitivity Function or the nonlinear responses to luminance and contrast.

At least, the need of these safety checks has to be included in the discussion because optimization on a visual task does not guarantee biological plausibility.

C. OTHER ANALYSIS

Previous comments imply that other analysis could have been performed in the final model:

- What are the features that fall within the same neighborhood?. Can you take one receptive field and show the receptive fields that fall in its neighborhood (or \lambda distance)?. Are these receptive fields similar?

- Can you give explicit evidences of the biological meaningfulness of the models with Divisive Normalization?

- The consideration of extra computations as Div. Norm. increases the number of paramters. Therefore, does the increase in the performance comes from the function of the new layer or just from the increased flexibility of the model?. Assessment of the accuracy of the model independent of the number of parameters should be performed.

D. LITERATURE (missing citations)

* There is literature on advantages of biological Divisive Normalization (with DN tuned according to biological goals). This literature, focused on biological interpretation of the parameters, assumes certain linear receptive fields (such as Gabor/Wavelet/DCT basis) and studies the advantages of representations with divisive normalization. In this context, the first paper to analyze the impact of Divisive Normalization on image classification (with fixed V1-like Gabors) was Coen-Cagli & Schwartz JoV 2013. Other papers focus on measuring the redundancy reduction ability of biological Divisive Normalization (e.g. Malo & Laparra Neural Computation 2010, Gomez-Villa et al. Journal of Neurophysiol. 2019). Even more meaningful than the redundancy is looking at the transmitted information from the input up to the internal representation (Malo J. Math. Neurosci. 2020). In all these cases, psychophysically meaningful divisive normalizations including Gaussian pooling over space and different frequency channels substantially reduce redundancy and maximize the transmitted information wrt versions with non-biological parameters.

* Other literature optimizes Divisive Normalization in deep learning context (as in this ICLR submission): e.g. DN for optimal rate-distortion image coding in Balle et al. ICLR 2017. Other examples that use a biological nonlinearity instead of or on top of the conventional nonlinearities in deep learning include Bertalmio et al. Sci. Rep. 2020 that generalize Wilson-Cowan layers (which are equivalent to Div.Norm.) and used these nonlinear layers in artificial architectures in a number of applications (classification, adversarial attacks, etc.).

E. MINOR

- The authors show the "first" receptive fields at some layer with and without normalization. However, how do they order the filters? what does it mean "first"?

- Analysis of RGB channels (as in Figs. 2-4) is not meaningful. In fact, quite soon in the pathway, opponent chromatic channels should appear and hence, achromatic, red-green and yellow-blue channels are more meaningful than RGB.


REFERENCES

[Balle17]  Johannes Ballé, Valero Laparra, Eero P. Simoncelli. End-to-end Optimized Image Compression. Int'l Conf on Learning Representations (ICLR), Apr 2017 2017

[Bertalmio20] Bertalmío, M., Gomez-Villa, A., Martín, A. et al. Evidence for the intrinsically nonlinear nature of receptive fields in vision. Sci Rep 10, 16277 (2020). https://doi.org/10.1038/s41598-020-73113-0

[Coen-Cagli13] Ruben Coen-Cagli, Odelia Schwartz; The impact on midlevel vision of statistically optimal divisive normalization in V1. Journal of Vision 2013;13(8):13. doi: https://doi.org/10.1167/13.8.13.

[Gomez-Villa19] Alexander Gomez-Villa, Marcelo Bertalmío, and Jesus Malo. Visual information flow in Wilson–Cowan networks. Journal of Neurophysiology 2020 123:6, 2249-2268

[Malo10] Jesús Malo, Valero Laparra; Psychophysically Tuned Divisive Normalization Approximately Factorizes the PDF of Natural Images. Neural Comput 2010; 22 (12): 3179–3206. doi: https://doi.org/10.1162/NECO_a_00046

[Malo20] J. Malo. Spatio-chromatic information available from different neural layers via Gaussianization. The Journal of Mathematical Neuroscience; Vol. 10, Iss. 1, (Dec 2020). DOI:10.1186/s13408-020-00095-8

[Martinez19] Martinez-Garcia M, Bertalmío M and Malo J (2019) In Praise of Artifice Reloaded: Caution With Natural Image Databases in Modeling Vision. Front. Neurosci. 13:8. doi: 10.3389/fnins.2019.0000

[Hepburn20] A. Hepburn, V. Laparra, J. Malo, R. McConville and R. Santos, "Perceptnet: A Human Visual System Inspired Neural Network For Estimating Perceptual Distance," 2020 IEEE International Conference on Image Processing (ICIP), 2020, pp. 121-125, doi: 10.1109/ICIP40778.2020.9190691.

[Webster11] Michael A. Webster; Adaptation and visual coding. Journal of Vision 2011;11(5):3. doi: https://doi.org/10.1167/11.5.3

[Zhang18] R. Zhang, P. Isola, A. Efros, E. Shechtman, and O. Wang, “The unreasonable effectiveness of deep features as a perceptual metric,” in Proceedings of the IEEE CVPR, 2018, pp. 586–595

**Summary Of The Paper:**

The manuscript 4523 studies the impact of a simplified Divisive Normalization in Alexnet in standard image classification tasks.
In particular the authors analyze the performance, the nature of the receptive fields, the geometry of the image class manifolds, and the sparsity of the responses.


**Summary Of The Review:**

The incorporation of biological nonlinearities in artificial networks is interesting both to enrich architectures in machine learning and to better study computational principles in neuroscience. Therefore, studies like this are (and will be) of interest for the ICLR audience.
I advocate for the publication of this work in the conference when the authors address/comment on the following concerns:
A. LIMITATIONS OF THE CONSIDERED NORMALIZATION AND TASK OBSCURE THE MESSAGE: clarify discussion!
B. IS THE FINAL DIVISIVE NORMALIZATION BIOLOGICALLY SENSIBLE?
C. OTHER ANALYSIS
D. LITERATURE (missing citations)

---

> ### Author Response · Authors · 2021-11-23
> **Part 1 of Response:**
>
> We thank the reviewer for their time spent reviewing our paper. We are encouraged that they found our work interesting for both ML and neuroscience, and of interest to the ICML audience. We incorporated their suggestions to our updated manuscript. Below we provide responses to the comments, in particular:
>
> $\\textbf{Part A and B}$
> $\\newline$
>
> The important points the reviewer raises in these sections are now briefly addressed in the discussion
> $\\newline$
>
> $\\textbf{Part C}$
> $\\newline$
> $\\textbf{“What are the features that fall within the same neighborhood?. Can you take one }$
> $\\textbf{receptive field and show the receptive fields that fall in its neighborhood}$
> $\\textbf{ (or lambda distance)?. Are these receptive fields similar?”}$
> -- Our illustrations of 16 RFs from layer 1 show adjacent RFs along the topological line of features, so the requested illustrations are shown there. $\\newline$
>
> -- In Fig. 27 of Appendix I we showed the correlation between RFs as a function of distance along the line. This shows that nearby features tend to be more weakly correlated than more distant features, and in many layers to be anticorrelated, in the divisive models. That is, the networks learn that RFs should be normalized by the responses of dissimilar, rather than similar, features. The figure also shows that, in several layers, divisive normalization can lead to lower correlation among all features, at all distances. We now briefly discuss these correlations in their own section of the paper ("Feature Correlations"). $\\newline$
>
> -- We also measured the mean of the abs. value of the circular difference in preferred orientations, as a function of distance. We found that this was indistinguishable from 45 deg at all distances, i.e. there was no sign that similarity or dissimilarity of preferred orientation played any systematic role in determining the RFs that normalize one another. $\\newline$
>
> -- It is interesting that this result -- both of anticorrelation or lowered correlation, and of a lack of involvement of preferred orientation -- is different from the finding of a study that learned divisive normalization to optimize the match of model output to V1 responses (Burg et al., 2021). Their model differed from ours in that there was only one convolutional layer, of 32 filters, followed by a single normalizing layer; and in that each neuron learned individual divisive weightings for all neighbors, whereas for us all neurons in a layer have the same learned neighborhood function.  They found that, among filters with a certain degree of orientation selectivity, the learned normalization weights more strongly connected RFs with similar orientations than with strongly dissimilar orientations. It would be interesting to explore the reasons for these different outcomes. $\\newline$
>
> $\\textbf{“Can you give explicit evidences of the biological meaningfulness of the models}$
> $\textbf{ with Divisive Normalization?”}$
>
> -- We know that DN is used extensively in cortical computations, and thus is biologically meaningful. This is a first exploration of what that may be good for computationally and in what contexts (e.g., combination with ‘canonical’ deep net normalizations). Eventually the field will learn how DN and other biological features combine to replicate biological representations, but at this stage we aim to learn how DN functions in terms of performance and the representations it builds in an otherwise vanilla deep net context.
>
> -- We did examine whether divisive normalization improves match to biology as measured by Brainscore, which measures consistency with the representations in the primate ventral visual stream. We examined the scores of the  models on the public (downloadable) set of data in Brainscore (although tests on the private data -- that which is used to evaluate web submissions of models -- may be more meaningful; we thus far have had technical problems in the web submission). Although it doesn’t appear that adding divisive norm consistently improves Brainscore relative to the same model without divisive norm, it is perhaps encouraging that the best performance on the three higher areas is given by a model with divisive norm.
>
> $\\textbf{See link for plots!}$
> LINK: https://imgur.com/a/5hCfn4M
> $\\textbf{“Does the increase in the performance comes from the function of the new layer or}$
> $\\textbf{ just from the increased flexibility of the model?”}$
> Overall, AlexNet trained on Imagenet has 62 million trainable parameters. Including Divisive normalization only adds 20 new parameters -- 4 parameters, lambda, alpha, beta and k, for each of the five layers in which DN is implemented -- to the 62 million of the AlexNet model. We find it highly unlikely that this negligible change in the number of parameters was the source of changed performance. For example, training a network with 20 weights omitted out of the 62 million parameters should have no noticeable effect on performance.

---

> > ### Author Response · Authors · 2021-11-23
> > **Part 2 of Response:**
> >
> > $\\textbf{Part D:}$
> > $\\textbf{The reviewer mentions additional literature on divisive normalization.}$
> > $\\newline$
> > We now refer to all of those sources in section 2.
> >
> > $\\textbf{Part E:}$
> > $\\textbf{“The authors show the "first" receptive fields at some layer with and }$
> > $\\textbf{without normalization. However, how do they order the filters? }$
> > $\\textbf{What does it mean "first"?”}$
> >
> > -- The receptive fields shown are those in the first layer. $\\newline$
> >
> > --The filters develop topologically on a line, numbered from 1 to N where N is the number of channels in a given layer, so 1 and N are the two ends of the line. The normalization is weighted by an exponential function of distance on this line, with learned space constant lambda. The first 16 are just the ones numbered 1 through 16. The main thing is that they are contiguous, hence at least the nearby filters among these 16 develop normalizing one another. We now mention this in the figure caption.
> > $\\newline$
> >
> > $\\textbf{The reviewer suggests to consider R-G, Y-B channels as opposed to RGB.}$ $\\newline$
> >
> > --We are not claiming a biological significance to viewing as RGB. Simply, the input images are in three colors, R, G and B, at each pixel, and the channels in the first layer are linear filters over all inputs, which can be most straightforwardly illustrated as filters for the R, G, and B channels. $\\newline$
> >
> > A final comment: we appreciate the extensive and illuminating discussion of the issues and the literature that the reviewer gives in Sections A,B,C. We feel badly that it's impossible to really engage those ideas in the tiny amounts of space we can give to them in the Introduction and Discussion, given all the other demands for the limited space. But at least, we do point to them.
> >
> > $\textbf{Citations for above discussion}$:
> >
> > Max F. Burg, Santiago A. Cadena, George H. Denfield, Edgar Y. Walker, Andreas S. Tolias, Matthias Bethge, and Alexander S. Ecker. Learning divisive normalization in primary visual cor- tex. bioRxiv, 2021. doi: 10.1101/767285. URL https://www.biorxiv.org/content/ early/2021/03/08/767285.

---

### Official Review · Reviewer_1XvT · 2021-11-02

**Correctness:** 3
**Technical Novelty And Significance:** 3
**Empirical Novelty And Significance:** 3
**Recommendation:** 6
**Confidence:** 4

**Main Review:**

Strengths:
- Divisive normalization is a crucial computation extensively studied by the neuroscience community; it is an important research question to compare and contrast this computation with the more recently proposed normalization techniques (batch / layer / group normalization) that work well with deep networks (DNNs).
- I find the writing to be clear and readable overall, I appreciate the clarity in describing the various normalization computations in Sec. 3.
- The authors show that adding divisive normalization in addition to another normalization technique (batch normalization) produces the best performing image classification model on two datasets (ImageNet, CIFAR-100)
- The authors perform an extensive set of visualization, representational, and Fourier analyses to understand why divisive normalization gives the above performance boost on image classification.
- It is interesting that as shown in Fig.2, the divisive normalization computation generally increases power in lower frequency bands. I believe divisive normalization performs a kind of smoothing operation with the exponential weighting that leads to increased sensitivity to lower frequency information.

Clarification requested:
(i) I would have been able to more confidently trust the ImageNet analyses had there been replication of the increased accuracy over multiple random seeds / initializations. In case this is time consuming on ImageNet, the authors could have used the ImageNet-100 dataset standardized in [1] which is smaller in size and faster to prototype.
(ii) Were the AlexNet parameters of all the 8 models compared in Fig.1 initialized to the same values? I could not find this information in the text, it would be good to add this information as it is important to ensure fair comparison of these architectures without any differences caused by a better AlexNet initialization that is independent of normalization.

Weaknesses:
- I don't think it is fair to conclude that divisive normalization produces first convolution layer kernels that look more like oriented Gabors while only comparing the first 16 kernels. Why did the authors only compare the first 16 kernels as opposed to looking at all 96 kernels? Despite this, I think the right comparison would be to quantitatively measure the orientation selectivity of the divisive and baseline models using sinusoidal grating stimuli for more concreteness in this analysis.
- The analyses conducted that suggest increased power in lower frequency spectrum, increased sparsity and manifold capacity are interesting, however, I am unable to coherently connect them to functional advantage (or disadvantage). The authors must better attempt to address why each of these findings are important and how they relate to the improved classification performance.
- Suggestion: Since the authors have ImageNet-trained models, it would be straightforward to evaluate whether the models with divisive normalization also perform better on out-of-distribution images (e.g. in [2]) to correlate some of the findings about low-frequency sensitivity to better generalization or shape-bias.
- Minor change: I would suggest using different markers for the various models in Figure 1 and 2 for better readability and accessibility.

References:
1. Tian, Yonglong, Dilip Krishnan, and Phillip Isola. "Contrastive multiview coding." Computer Vision–ECCV 2020: 16th European Conference, Glasgow, UK, August 23–28, 2020, Proceedings, Part XI 16. Springer International Publishing, 2020.
2. Geirhos, R., Temme, C. R. M., Rauber, J., Schütt, H. H., Bethge, M., & Wichmann, F. A. (2018). Generalisation in humans and deep neural networks. arXiv preprint arXiv:1808.08750.

**Summary Of The Paper:**

The authors study the role of the biologically realistic divisive normalization computation in the context of deep learning models trained to perform image classification tasks. The authors compare divisive normalization (scaling neuronal response by exponentially weighted sum of its neighbors) with those normalization techniques commonly used in machine learning models (such as batch normalization, layer normalization etc) and find that their implementation of divisive normalization provides improved image recognition performance. The authors perform a range of analyses to representationally understand why divisive normalization provides the above-mentioned increase in classification accuracy.

**Summary Of The Review:**

I find that this paper presents promising evidence suggesting that divisive normalization in combination with batch normalization can improve classification performance. The authors attempt to explain the performance gain by analyzing the representation of divisive normalization model in comparison to baseline models without normalization (or with other types of normalization). The findings are interesting but I am finding it difficult to grasp the main takeaway from the paper, especially since the gains produced by divisive normalization are quite small. I do not recommend acceptance of this paper at the current stage, but I am willing to change my decision if my comments above are addressed by the authors during the rebuttal phase.

Key factors I am concerned about that can influence my score:
1. Answers to the clarification requested in the main review above.
2. More clarity on connecting the representational analyses of divisive normalization models and other baselines with functional advantages (such as the performance gain on ImageNet and CIFAR-100).
3. I would be very impressed if the divisive (or divisive batch) models perform significantly better than other baselines on the OOD generalization benchmark referred in my main review (or any other comparable OOD generalization task that the authors find relevant).

======== UPDATE AFTER AUTHOR RESPONSE ==========
I am increasing my score to 6 and am recommending acceptance of this paper. I am updating my initial opinion of this paper, in my opinion the paper performs interesting experiments and analyses on OOD generalization and feature selectivity benefits produced by divisive normalized networks. I am not giving the paper an even higher score since I feel that these benefits / gains are quite small compared to just using batch normalization.

---

> ### Author Response · Authors · 2021-11-23
> **Part 1 of Response:**
>
> We thank the reviewer for their time spent reviewing our paper; we are pleased to hear that they found our work presents promising evidence and has interesting findings. We incorporated their suggestions to our updated manuscript. Below we provide responses to the comments, in particular:
>
> $\\textbf{Two Clarifications:}$
> $\textbf{(i)  “ I would have been able to more confidently trust the ImageNet analyses had there}$
> $\textbf{  been replication of the increased accuracy over multiple random seeds / initializations.”}$
>
> --We did do different runs for ImageNet, but they varied very little, with final performance differing by .1 - .2 points. We therefore did not find it worth the time and computational resources to do a full series of 30 seeds on Imagenet. We now mention this in the paper.
>
> $\textbf{(ii)     “Were the AlexNet parameters of all the 8 models compared in Fig.1 initialized to}$
> $\textbf{the same values?”}$
>
> --The weights for each convolutional layer in all 8 models were initialized using the same statistics but different seeds. We used the Kaiming He initialization, with weights initialized from a Gaussian Distribution of mean 0 and standard deviation proportional to the square root of 1/(number of inputs to that neuron). This is discussed in Appendix A. As noted in the answer to point (i), changing seeds caused only very small variations in performance (0.1 - 0.2 points).
>
> $\\textbf{Discussing Weakness and Changes:}$
> $\textbf{“Why did the authors only compare the first 16 kernels as opposed to looking at }$
> $\textbf{ all 96 kernels?” The reviewer also suggests to quantitatively measure}$
> $\textbf{ orientation selectivity.}$
>
> $\\newline$
> -- For the sake of practicality, we only illustrate the first 16 kernels in the paper since including all 96 kernels simply takes too much space. However, the first 16 kernels are entirely representative.
> $\\newline$
>
> -- In response to the reviewer, we have now measured the orientation selectivity of each filter in each color channel for the models. We find that, for the Batch and Group models, addition of divisive normalization does increase the orientation selectivity of the layer 1 units: We compute an orientation tuning curve by taking the Fourier transform of each 11x11 filter, dividing the 2-D wavenumbers k into 12 orientation bins (e.g., if the vector k points between 0 and 15 deg or between 180 and 195 deg, it goes into the bin of orientation 0 to 15 deg), and calling the response to a given orientation bin the maximum of the amplitude of the Fourier transform for k’s within that bin. This is equivalent to the maximum response to a full-RF sinusoidal grating of any spatial frequency and phase within the given orientation bin. We calculated the circular variance, CV = 1-F1/F0, where F1 and F0 are the first harmonic and DC of the orientation tuning curve. CV=1 represents no orientation selectivity, while CV=0 is maximum orientation selectivity (nonzero response only to a single orientation). We now include this material in Appendix K and reference it in the main text.
> $\\newline$
>
> -- We plot the distributions of the CV values for all 3*96 features, first for the Batch and DivisiveBatch models, then for the NoNorm and Divisive models. We use 50 bins and the curves fitted are plotted using the scipy B-spline fit. It appears the DivisiveBatch is shifted left (higher orientation selectivity) compared to the Batch.
> $\\newline$
>
> -- In the link, we also include the boxplots for the same four distributions above, which suggest a modest drop in the medians and interquartile ranges of the divisively normalized distributions, particularly the DivisiveBatch relative to Batch:
>
> $\\newline $
> -- $\\textbf{Link to plots!}$: https://imgur.com/a/NujGleE $\\newline$
>
> -- More to follow discussing the distribution of circular variance across the receptive fields

---

> > ### Author Response · Authors · 2021-11-23
> > **Part 2 of Response:**
> >
> > -- To determine if the CV distributions differ significantly, we use the KS (Kolmogorov-Smirnov) 2 sample test. Specifically, it determines whether two sets of data come from the same distribution. We find that the Batch vs. Divisive Batch distributions are significantly different, as are the Group vs. DivisiveGroup.
> > $\\newline$
> >
> > $\\textbf{KS Test } $
> >
> > $ \\begin{array}{|c|c|}
> > \\text{Model} & \\text{p-value} \\newline
> > \\text{NoNorm vs Divisive} & .151 \\newline
> > \hline
> > \\text{Batch vs DivisiveBatch} & .000197 \\newline
> > \hline
> > \\text{Group vs DivisiveGroup} & .0012  \\newline
> > \hline
> > \\text{Layer vs DivisiveLayer} & .182
> > \\end{array} $
> > $\\newline$
> >
> > -- Next, we use the MannWhitney U test, which determines whether samples from one distribution tend to be significantly larger than samples from the other distribution. Again, Batch vs. DivisiveBatch and Group vs. Divisive Group are significantly different. Since in both cases the model with Divisive normalization has the lower median (medians: Batch, .364; DivisiveBatch, .332; Group, .369; DivisiveGroup, .336), we take this to mean that in these two cases the addition of Divisive normalization increases orientation selectivity. $\\newline$
> >
> > $\\textbf{MannWhitney Test}$
> > $\\newline$
> >
> > $ \\begin{array}{|c|c|}
> > \\text{Model} & \\text{p-value} \\newline
> > \\text{NoNorm vs Divisive} & .186 \\newline
> > \hline
> > \\text{Batch vs DivisiveBatch} & .0016 \\newline
> > \hline
> > \\text{Group vs DivisiveGroup} & .0051  \\newline
> > \hline
> > \\text{Layer vs DivisiveLayer} & .135
> > \\end{array} $
> > $\\newline$
> >
> > --We also calculated the skew for each distribution. A negative skew implies there is more bulk on the right and a long tail on the left, a positive value implies the opposite.  Thus a positive skew suggests a greater bulk of units with low CV (higher orientation selectivity). For Batch and Group, addition of Divisive normalization converted a negative skew to a positive skew, again consistent with Divisive normalization causing a lowering of CV’s for these two models.
> > $\\newline$
> >
> > $ \\begin{array}{|c|c|}
> > \\text{\\textbf{Model}} & \\textbf{Skew} \\newline
> > \\text{NoNorm} & .124 \\newline
> > \hline
> > \\text{Divisive} & .271 \\newline
> > \hline
> > \\text{Batch} & -.0499 \\newline
> > \hline
> > \\text{DivisiveBatch} & .2064 \\newline
> > \hline
> > \\text{Group} & -.089  \\newline
> > \hline
> > \\text{DivisiveGroup} & .304  \\newline
> > \hline
> > \\text{Layer} & .1506 \\newline
> > \hline
> > \\text{DivisiveLayer} & .039\\newline
> > \\end{array} $
> >
> > Again, this material is presented in Appendix K and referred to in the text.
> >
> > $\\newline$
> > $\\textbf{“The authors must better attempt to address why each of these findings are }$
> > $\\textbf{ important and how they relate to the improved classification performance.”}$
> > $\\newline$
> >
> > -- The question of what aspects of the neural representations account for the performance of deep nets remains a critical and unsolved problem in the field. We are not going to solve that problem in this paper. However, uncovering aspects of the representations that are changed in ways that correlate with performance (sparsity, power in lower-frequency spectrum) suggests hypotheses as to causal representational features that can contribute to the field’s continuing investigation of this question. Furthermore the manifold capacity and the associated geometric features that contribute to it (manifold radius, dimension, and center correlation) are direct measures of network capacity and thus theoretically provide real insight into performance. Our finding that, with divisive normalization, manifold capacity is actually reduced in intermediate layers yet increased at the end is surprising and raises questions for the field as to how a reduced intermediate capacity can contribute to a final increased capacity and improvement in performance. We try to better address this in the introduction/discussion. $\\newline$
> >
> > $\textbf{“Minor change: Use different markers for the various models in Figure 1 and 2 }$
> > $\\textbf{for better readability and accessibility.”}$
> >
> > We made the necessary changes.
> >
> > $\textbf{“More clarity on connecting the representational analyses of divisive }$
> > $\\textbf{normalization models and other baselines with functional advantages”}$
> >
> > Please see our answer above re functional advantages.

---

> > > ### Author Response · Authors · 2021-11-23
> > > **Part 3 of Response:**
> > >
> > > $\textbf{The reviewer suggests to evaluate the models on out-of-distribution images to correlate}$
> > > $\textbf{ low frequency sensitivity to shape bias.}$
> > >
> > > -- We have found indications of divisive normalization giving better OOD generalization, which we now discuss in a short section "Out of Distribution (OOD) Images" right before the Discussion, as follows:  $\\newline$
> > >
> > > --$\textbf{ Adversarial Attacks}$: In Appendix H of the paper, we addressed some initial explorations of testing the robustness of the models using adversarial attacks. For white box attacks (PGD and FGSM), which exploit the particular structure of the network and its gradients, most of the normalized models did not perform well relative to the NoNorm model (except for Layer, Batch and Divisivegroup). However, a test of OOD performance is given by black box attacks, which use OOD images without knowledge of the network processing them. Against a black box attack (L2 Additive Gaussian noise), DivisiveGroup performed best against moderate-strength attacks, while DivisiveBatch and Divisive Layer performed best against the strongest attacks we tried (see Figure 26 in Appendix H). This is preliminary evidence of divisive norm improving OOD performance. $\\newline$
> > >
> > > --$\\textbf{ Shape vs texture bias} $: We studied shape vs texture bias using the texture-vs-shape package on github. This dataset has 16 shape classes, each with 80 photos, with textures substituted from images of the other classes,  for a total dataset of 1280 images. These are a subset of the textures and images used in the stylized ImageNet dataset. For each image, when a model’s inferred category matched the category of either the shape or the texture used in the decision, this was counted as a shape or a texture decision, respectively. The shape bias is the number of shape decisions, divided by the sum of shape and texture decisions. We found that the batch-norm model had a stronger shape bias than no-norm, and, most importantly, that addition of divisive norm to either of these models increased the shape bias (see figure; vertical lines show the mean of model represented with corresponding color).  $\\newline$
> > >
> > > -- However, many of the models’ decisions were “other” -- a category corresponding to neither the shape nor the texture -- and the models with divisive normalization had increased “other” decisions (66.8% for Divisive vs. 61.1% for NoNorm; 72.67% for DivisiveBatch vs. 63.75% for Batch). Thus, the models with divisive normalization had an increased shape bias, yet had the same number of or fewer shape decisions out of all categorizations. This gives some hope of divisive normalization and its increased low- or moderate-spatial-frequency power improving shape sensitivity, as the reviewer suggests, but does not allow clear conclusions. $\\newline$
> > >
> > > -- $\\textbf{See link for plots!}$
> > > A link to the plot: https://imgur.com/a/Wf1UHtc  $\\newline$
> > >
> > > $\\textbf{Citations for above discussion:}$
> > >
> > > Ian J. Goodfellow, Jonathon Shlens, and Christian Szegedy. Explaining and harnessing adversarial examples, 2015.
> > >
> > > Robert Geirhos, Patricia Rubisch, Claudio Michaelis, Matthias Bethge, Felix A. Wichmann, and Wieland Brendel. Imagenet-trained cnns are biased towards texture; increasing shape bias improves accuracy and robustness. CoRR, abs/1811.12231, 2018. URL http://arxiv.org/abs/ 1811.12231.

---

> > > > ### Comment · Reviewer_1XvT · 2021-11-29
> > > > **Update after reading author response**
> > > >
> > > > I thank the authors for such detailed and elaborate response to my queries and that of the other reviewers. I believe the paper is in much better shape after these additional analyses.
> > > >
> > > > Re. "For the sake of practicality, we only illustrate the first 16 kernels in the paper since including all 96 kernels simply takes too much space", I still believe it would be good to have a more principled way to select these kernels rather than selecting the first 16.
> > > >
> > > > Albeit a very small difference (referring to the box plot), it kind of looks like adding divisive normalization is increasing orientation selectivity.
> > > >
> > > > I am particularly very interested in the OOD generalization analysis. It does look like Divisive + Batch is doing better than just Batch in several instances, which is great and needs further investigation as to why divisive normalization helps in better OOD generalization.
> > > >
> > > > Overall, I think the paper has improved since the first submission, and I am happy to increase my score to 6. I believe this paper could be accepted as the results look promising and it will spark interesting discussions and future work at the ICLR meeting.

---

### Decision · Program_Chairs · 2022-01-20

**Decision:**

Accept (Poster)

**Comment:**

This paper explores addition of a version of divisive normalization to AlexNets and
compares performance and other measures of these networks to those
with more commmonly used normalization schemes (batch, group, and
layer norm).  Various tests are performed to explore the effect of
their divisive normalization.

Scores were initially mixed but after clarifications for design and
experiment decisions, and experiments run in response to comments by
the reviewers the paper improved significantly.  While reviewers still
had several suggestions for further improvements, after the authors'
revisions reviewers were in favor of acceptance which I support.